# Cancer cell-intrinsic biosynthesis of itaconate promotes tumor immunogenicity

Zining Wang [1,5], Lei Cui[1,5], Yanxun Lin[1], Bitao Huo[1,2], Hongxia Zhang[1], Chunyuan Xie [1],
Huanling Zhang[1], Yongxiang Liu[1], Huan Jin[1], Hui Guo[1], Mengyun Li[1,3], Xiaojuan Wang[1], Penghui Zhou [1],
Peng Huang [1,2], Jinyun Liu [1,2,4] & Xiaojun Xia [1,4✉]

## Abstract

**The Krebs cycle byproduct itaconate has recently emerged as an important metabolite regulating macrophage immune functions, but its role in tumor cells remains unknown. Here, we show that increased tumor-intrinsic cis-aconitate decarboxylase (ACOD1 or CAD, encoded by immune-responsive gene 1, *Irg1*) expression and itaconate production promote tumor immunogenicity and anti-tumor immune responses. Furthermore, we identify thimerosal, a vaccine preservative, as a specific inducer of IRG1 expression in tumor cells but not in macrophages, thereby enhancing tumor immunogenicity. Mechanistically, thimerosal induces itaconate production through a ROS-RIPK3-IRF1 signaling axis in tumor cells. Further, increased IRG1/itaconate upregulates antigen presentation-related gene expression via promoting TFEB nuclear translocation. Intratumoral injection of thimerosal induced itaconate production, activated the tumor immune microenvironment, and inhibited tumor growth in a T cell-dependent manner. Importantly, IRG1 deficiency markedly impaired tumor response to thimerosal treatment. Furthermore, itaconate induction by thimerosal potentiates the anti-tumor efficacy of adoptive T-cell therapy and anti-PD1 therapy in a mouse lymphoma model. Hence, our findings identify a new role for tumor intrinsic IRG1/itaconate in promoting tumor immunogenicity and provide a translational means to increase immunotherapy efficacy.**

**Keywords** Itaconate; Immunogenicity; Thimerosal; Immunotherapy
**Subject Categories** Cancer; Immunology; Metabolism

## Introduction

Cancer immunotherapy has become the major theme of cancer treatment regimens in the past few years, testifying to the genuine capability of the immune control of cancer (Kraehenbuehl et al, 2022; Waldman et al, 2020). Immune checkpoint inhibitors and adoptive T cell transfer therapies have demonstrated impressive efficacy across several different cancer types (Ansell et al, 2015; Garon et al, 2015). Despite the prevalent success in treating multiple types of cancers, the response rate to immunotherapy is quite low (Galluzzi et al, 2018; Zou et al, 2016). The immunogenicity of tumors refers to the ability of tumor cells to be recognized by the immune system, initiating the "cancer-immunity" cycle. Accumulating evidence suggests that the efficacy of cancer immunotherapy is closely linked to tumor immunogenicity and the tumor microenvironment (Ayers et al, 2017; Bruni et al, 2020; Rizvi et al, 2015; Turajlic et al, 2017). Increasing tumor immunogenicity has the potential to enhance the anti-tumor immune response and transform the immune "cold" tumor microenvironment into a more favorable "hot" environment, ultimately improving therapy outcomes and broadening the population that may benefit from immunotherapy (Pfirschke et al, 2016; Wang et al, 2019). Hence, there is an urgent need to identify innovative therapeutic approaches that can enhance the immunogenicity of tumors and augment their responsiveness to immunotherapy.

Itaconate, a tricarboxylic acid cycle (TCA) byproduct derived from *cis*-aconitate by the enzyme *cis*-aconitate decarboxylase (ACOD1), which is encoded by immune-responsive gene 1 (*Irg1*), acts as a critical metabolite in antimicrobial and anti-inflammatory responses in macrophages (Chen et al, 2022; Chen et al, 2020; Michelucci et al, 2013; Mills et al, 2018; Runtsch et al, 2022; Zhang et al, 2022). Itaconate production is induced in macrophages by various factors such as LPS, bacterial infection, tumor cell co-culture, and potentially by the ZIKA virus via a ZBP1-RIPK3 axis in neurons (Daniels et al, 2019; Michelucci et al, 2013; Weiss et al, 2018). In a peritoneal tumor model, the tumor induces a high level of itaconate in macrophages to up-regulate oxidative phosphorylation (OXPHOS) and mitochondrial reactive oxygen species (ROS), thereby activating mitogen-activated protein kinase (MAPK) signaling in tumor cells to promote tumor growth (Weiss et al, 2018). Recent studies also demonstrated that itaconate derived from myeloid-derived suppressor cells (MDSCs) promoted tumor growth by inhibiting T cell cytotoxic function or inducing CD8 + T

[1]State Key Laboratory of Oncology in South China, Guangdong Provincial Clinical Research Center for Cancer, Sun Yat-sen University Cancer Center, Guangzhou, China.
[2]Metabolic Innovation Center, Zhongshan School of Medicine, Sun Yat-sen University, Guangzhou, China. [3]MOE Key Laboratory of Gene Function and Regulation, State Key Laboratory of Biocontrol, School of Life Sciences, Sun Yat-sen University, Guangzhou, China. [4]Hainan Academy of Medical Sciences, Hainan Medical University, Haikou, China.
[5]These authors contributed equally: Zining Wang, Lei Cui. ✉E-mail: xiaxj@sysucc.org.cn

cell exhaustion in hepatocellular carcinoma (Gu et al, 2023; Zhao et al, 2022). Additionally, itaconate has been found to impair the macrophage polarization into M1 subtype by targeting TET2 enzyme activity, thereby creating an immunosuppressive tumor microenvironment (Chen et al, 2023). These findings underscore the significant immune-modulating role of itaconate in macrophages. However, earlier studies indicate that tumor cells typically exhibit undetectable or very low levels of itaconate production (Chen et al, 2023; Weiss et al, 2018), and the impact of itaconate on tumor cell biology remains largely unexplored. The potential for tumor cells to produce itaconate and the impact of tumor-intrinsic itaconate on the regulation of anti-tumor immunity has remained largely unclear.

Here we demonstrate that the itaconate could be induced in tumor cells, leading to enhanced tumor immunogenicity and anti-tumor immune responses. Based on IRG1 protein expression induction screening, we find that thimerosal, a widely used vaccine preservative, can significantly induce IRG1 expression and itaconate production in tumor cells, but not macrophages. Thimerosal significantly induced tumor-intrinsic IRG1 expression and itaconate production to enhance tumor immunogenicity. Mechanistically, itaconate upregulates TFEB-mediated antigen presentation and subsequently elevates tumor immunogenicity, which is attenuated by TFEB knockdown. Specifically, thimerosal activates the ROS-RIPK3-IRF1 signaling axis to induce IRG1 expression and itaconate production in tumor cells. In a mouse tumor model, intratumoral injection of thimerosal suppresses tumor growth in a T cell-dependent manner and promotes activation of the tumor immune microenvironment. Tumors lacking IRG1 exhibit diminished response to thimerosal treatment. Furthermore, thimerosal enhances the in vivo effectiveness of adoptive T cell therapy and anti-PD1 treatment in a mouse lymphoma model. Our findings uncovered a new role for tumor-intrinsic itaconate production in promoting tumor immunogenicity and anti-tumor immune response. Additionally, we have identified thimerosal as a specific inducer of tumor-intrinsic IRG1/itaconate, offering a promising strategy for improving the effectiveness of anti-tumor immunotherapy.

## Results

### Itaconate triggers tumor immunogenicity

To test whether itaconate could regulate tumor immunogenicity, we adapted an in vitro tumor-T cell co-culture assay system and measured tumor cell-elicited T cell activation which reflects tumor cell immunogenicity. We pre-treated EG7 cells (a mouse lymphoma cell line expressing chicken ovalbumin (OVA)) with itaconate derivatives dimethyl itaconate (DI) or 4-octyl itaconate (4OI) (which were cell-permeable) for 16 h, then co-cultured them with OVA-specific CD8$^+$ T cell hybridoma B3Z cells or primary OT-I T cells. After 24 h, IFNγ production level or IL-2 promoter–driven LacZ activity was measured as surrogate markers for T cell activation. The results show that EG7 cell treated with DI or 4-OI could significantly increase IFNγ production and LacZ activity in the tumor-T cell co-culture system (Fig. 1A,B). In addition, the treatment of itaconate plus digitonin (digitonin could help itaconate entry into cells) but not itaconate itself in EG7 could

also increase the LacZ activity in the tumor-T cell co-culture system (Fig. 1C, Appendix Fig. S1A,B). These results suggest that itaconate treatment on tumor cells could trigger tumor immunogenicity. A similar result was observed in B16-OVA cells pre-treated with 4-OI (Fig. 1D). In addition, the itaconate derivatives-treated tumor cells also increased the intracellular GZMB and IFNγ expression, and CD25 expression on T cells in the tumor-T cell co-culture system (Fig. 1E; Appendix Fig. S1C,D). Moreover, the itaconate derivatives and itaconate plus digitonin also enhanced the expression of MHC-I and MHC-I SIINFEKL expression on tumor cells (Fig. 1F,G; Appendix Fig. S1E,F). Overall, these data suggested that itaconate could trigger tumor immunogenicity.

### Thimerosal induces tumor-intrinsic itaconate production for tumor immunogenicity

Previous studies have frequently used itaconate derivatives to mimic the function of endogenous itaconate, such as anti-inflammatory or anti-bacterial (Hooftman et al, 2020; Lampropoulou et al, 2016; Mills et al, 2018; Runtsch et al, 2022). While, recent reports showed large amounts of itaconate can be induced in macrophages and suppress the anti-tumor immune response (Chen et al, 2023; Zhao et al, 2022). But whether endogenous itaconate production can be induced in tumor cells and in turn trigger tumor immunogenicity is unknown. Thus, we screened a small library of FDA-approved drugs (Appendix Table S1, a shortlist of pre-selected drugs that induce cell death or proliferation inhibition in our previous study (Wang et al, 2019)) for their effects on inducing expression of IRG1, the key enzyme responsible for itaconate production. We identified that thimerosal, a commonly used vaccine preservative, could significantly induce IRG1 expression in tumor cells (Fig. 2A; Appendix Fig. S2A–C). Different from macrophages, the IRG1 expression in tumor cells could be induced by thimerosal but not LPS (Fig. 2B). On the contrary, thimerosal could only induce the IRG1 expression in tumor cells but not macrophages, indicating different induction mechanisms of IRG1 expression in different cells (Fig. 2C; Appendix Fig. S2D). Furthermore, metabolomics screening shows thimerosal influences multiple metabolic production, while itaconate was increased significantly (Appendix Fig. S2E; Dataset EV1). Consistently, thimerosal could only induce the itaconate production in EG7 cells but not macrophages (Appendix Fig. S2F). Moreover, thimerosal induced IRG1 mRNA expression and intracellular itaconate production but no other metabolites in the TCA cycle in tumor cells (Fig. 2D,E; Appendix Fig. S2G). In line with tumor immunogenicity induced by itaconate, thimerosal also triggers tumor immunogenicity evidenced by significantly increased IL-2 promoter–driven LacZ activity and IFNγ production in T cells co-cultured with thimerosal-treated tumor cells (Fig. 2F,G; Appendix Fig. S2H). Consistently, OT-I T cells co-cultured with thimerosal-treated tumor cells also upregulated the expression levels of activation markers CD25, CD69, and effector molecules such as intracellular Granzyme B (GZMB) and IFNγ (Fig. 2H,I; Appendix Fig. S2I). Thimerosal also enhanced the expression of MHC-I SIINFEKL expression on tumor cells (Appendix Fig. S2J). We next treat tumor cells with thimerosal in combination with 4-OI. The results show that the combination treatment did not further increase tumor immunogenicity, suggesting that itaconate induction mediates thimerosal effect

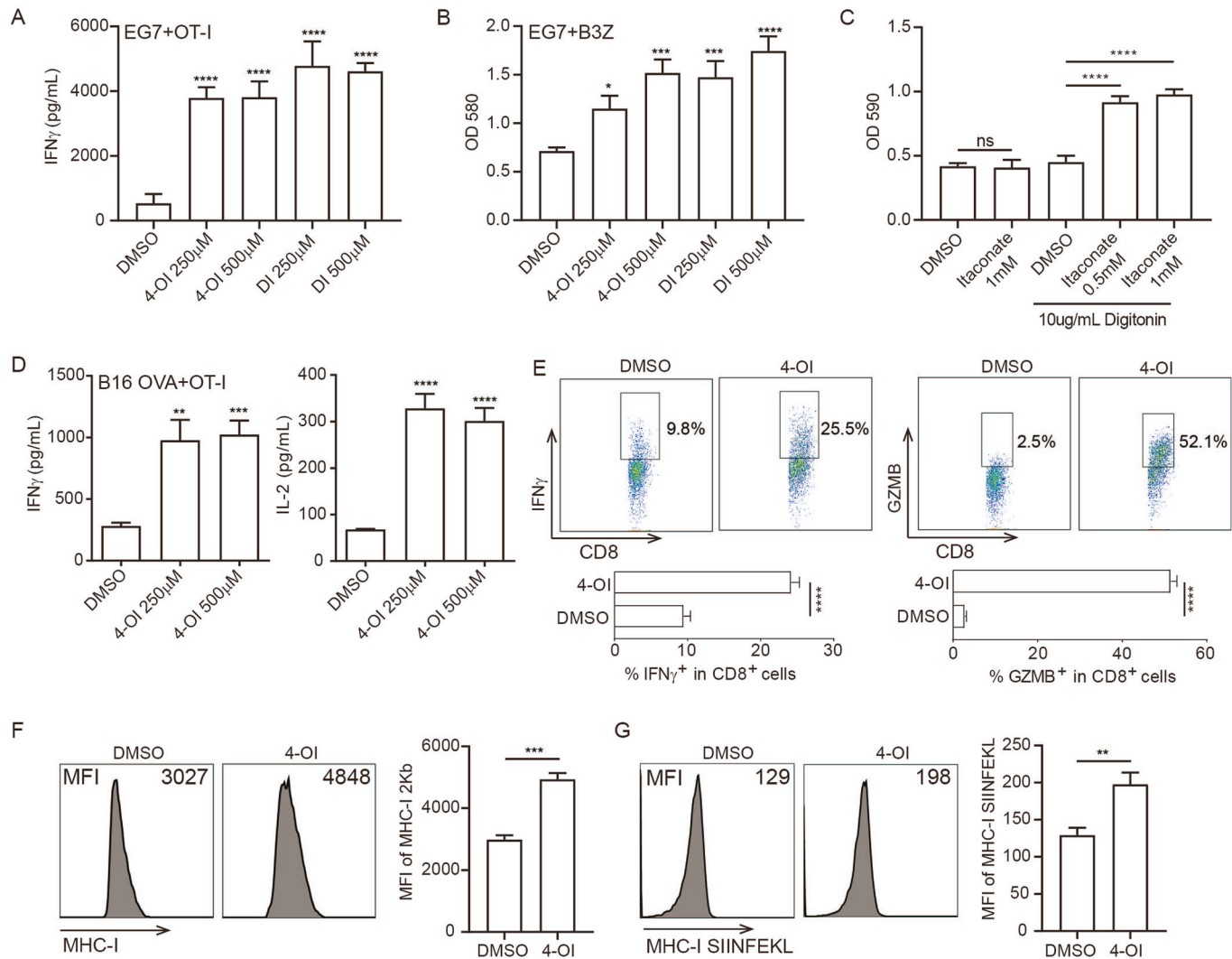

**Figure 1. Itaconate promotes tumor immunogenicity.**

(A, B) EG7 cells were treated with 4OI, DI, or itaconate for 16 h, followed by co-culture with OT-I or B3Z for an additional 24 h, and then the LacZ activity and supernatant IFNγ levels were measured. ****$P < 0.0001$ (A), *$P = 0.0243$, ***$P = 0.0002$, ***$P = 0.0004$, ****$P < 0.0001$ (B) in one-way ANOVA analysis of variance with Bonferroni's post-test. (C) EG7 cells were treated with itaconate or itaconate plus digitonin for 16 h, then co-culture with B3Z cells for an additional 24 h, then the LacZ activity was measured. ****$P < 0.0001$ in one-way ANOVA analysis of variance with Bonferroni's post-test. (D) B16-OVA cells were treated with 4OI for 16 h, followed by co-culture with OT-I for an additional 24 h, then the supernatant IFNγ or IL2 levels were measured. **$P = 0.001$, ***$P = 0.0007$ (left panel), ****$P < 0.0001$ (right panel) in one-way ANOVA analysis of variance with Bonferroni's post-test. (E). EG7 cells were treated as described in (A), then the intracellular expression levels of GZMB and IFNγ in OT-I cells were measured by FACS. ****$P < 0.0001$ in Student's $t$ test. (F, G) EG7 cells were treated with 4OI (500 μM) for 16 h, then the expression levels of MHC-I and MHC-I SIINFEKL were detected by FACS. ***$P = 0.0002$ (F), **$P = 0.0033$ (G), in Student's $t$ test. The graph is shown as mean ± SD of $n = 3$ for all panels. Source data are available online for this figure.

(Fig. 2J; Appendix Fig. S2K). Conversely, IRG1 knockout (KO) in EG7 tumor cells via the CRISPR/Cas9 technique not only abolished itaconate production induced by thimerosal treatment (Fig. 2K,L; Appendix Fig. S2L–N) but also markedly attenuated the IFNγ production and IL-2 promoter-driven LacZ activity in T cells co-cultured with EG7 cells (Fig. 2M; Appendix Fig. S2O). Moreover, IRG1 KO also abolished the MHC-I and MHC-I SIINFEKL expression enhanced by thimerosal (Fig. 2N; Appendix Fig. S2P). Taken together, these data identified thimerosal as a tumor-specific itaconate inducer, which could trigger tumor immunogenicity by inducing IRG1 expression and itaconate production in tumor cells.

## Itaconate induces tumor immunogenicity by upregulating TFEB-directed antigen presentation

To investigate how itaconate enhanced tumor immunogenicity, we performed RNA-seq analysis to identify different gene expression patterns in EG7 cells after 4-Ol treatment. Kyoto Encyclopedia of Genes and Genomes (KEGG) analysis found that the lysosome biogenesis genes were significantly upregulated by 4-OI treatment (Appendix Fig. S3A,B), and these upregulated genes can also be verified by qPCR or Western blot (Appendix Fig. S3C,D), which is consistent with the recent report showing

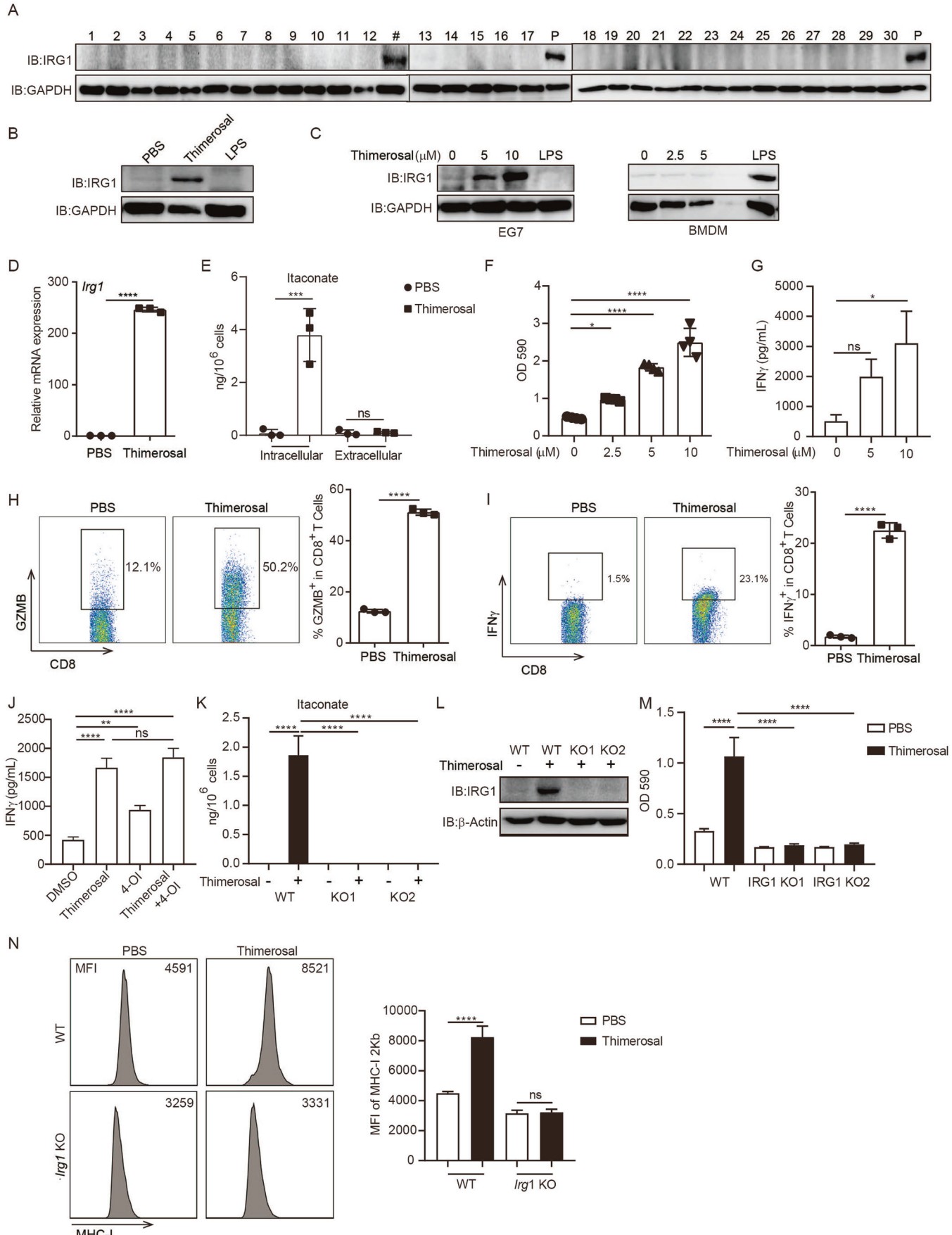

◄   **Figure 2.   Thimerosal specifically induces IRG1/itaconate in tumor cells.**

(A) EG7 cells were treated with indicated drugs (5 μM, Appendix Table S1) for 16 h, then the IRG1 expression was detected by WB; P: positive control using whole cell lysate of 293 cells overexpressing IRG1, # is thimerosal. (B) EG7 cells were treated with thimerosal (10 μM) or LPS (100 ng/mL) for 16 h, then the IRG1 expression was detected by WB. (C) EG7 or BMDM cells were treated as in (B), then the IRG1 expression was detected by WB. (D, E) EG7 cells were treated with thimerosal (10 μM) for 16 h, then the mRNA expression level of IRG1 was measured by qPCR, and the production of itaconate in intracellular and extracellular was measured by HPLC. ****$P < 0.0001$ (D), in Student's $t$ test, ***$P = 0.0001$ (E) in one-way ANOVA analysis of variance with Bonferroni's post-test. (F, G) EG7 cells were treated with thimerosal (10 μM) for 16 h, then co-cultured with B3Z cells or OT-I for an additional 24 h, and then the LacZ activity or supernatant IFNγ levels were measured. *$P = 0.0217$, ****$P < 0.0001$ (F), *$P = 0.0128$ (G) in one-way ANOVA analysis of variance with Bonferroni's post-test. (H, I) EG7 cells were treated as in (C), then co-cultured with OT-I cells for an additional 24 h, then the intracellular expression levels of GZMB and IFNγ were measured by FACS. ****$P < 0.0001$ (H, I) in Student's $t$ test. (J) EG7 cells were treated with 4OI (250 μM) or combined with thimerosal (10 μM), then co-cultured with OT-I for an additional 24 h, and then the supernatant IFNγ was measured by ELISA. ****$P < 0.0001$, **$P = 0.005$, in one-way ANOVA analysis of variance with Bonferroni's post-test. (K, L) EG7 WT or *Irg1* KO cells were treated with thimerosal (10 μM) for 18 h, then the production of itaconate was measured by HPLC, and the protein expression levels of IRG1 were detected by WB. ****$P < 0.0001$ in one-way ANOVA analysis of variance with Bonferroni's post-test. (M) EG7 WT or *Irg1* KO cells were treated with thimerosal (10 μM) for 16 h, followed by co-culture with B3Z for another 24 h, and then the LacZ activity was measured. ****$P < 0.0001$ in one-way ANOVA analysis of variance with Bonferroni's post-test. (N) EG7 WT or *Irg1* KO cells were treated with thimerosal (10 μM) for 24 h, and then the expression levels of MHC-I were detected by FACS. ****$P < 0.0001$ in one-way ANOVA analysis of variance with Bonferroni's post-test. The graph is shown as mean ± SD of $n = 4$ for (F) and $n = 3$ for other panels. Source data are available online for this figure.

lysosome biogenesis induction by itaconate in macrophages (Zhang et al, 2022). TFEB is known as a master transcription factor for lysosome biogenesis, which could be modified by itaconate in macrophages (Sardiello et al, 2009; Settembre et al, 2011; Zhang et al, 2022). A previous report shows that overexpression of TFEB in tumor cells could up-regulate the HLA-ABC expression and consequent tumor antigenicity (Lee et al, 2022). This prompted us to investigate whether the 4-OI induces tumor immunogenicity depending on TFEB. Firstly, we used the 4-OI, thimerosal, or itaconate to treat cells and found that 4-OI/thimerosal/itaconate treatment significantly increased TFEB translocation to nuclear (Fig. 3A; Appendix Fig. S3E). While KO IRG1 significantly attenuates thimerosal-inducing TFEB translocation to nuclear (Appendix Fig. S3F). Mutation of the C270 in TFEB also attenuates its nuclear translocation induced by thimerosal (Appendix Fig. S3G). Next, we found that 4-OI and thimerosal treatment increased the antigen presentation gene expressions such as *B2m*, *Tap1*, and *Tap2* (Fig. 3B). Consistently, Chromatin immunoprecipitation followed by qPCR (ChIP-qPCR) result showed that the thimerosal or 4-OI treatment also enhanced TFEB binding to the *B2m*, *Tap1* and *Tap2* gene promoter region (Fig. 3C). Moreover, the knockdown of TFEB in tumor cells via shRNA not only decreased the mRNA levels of the lysosome biogenesis genes but also levels of these antigen presentation genes induced by 4-OI and thimerosal treatment (Fig. 3D,E; Appendix Fig. S3H,I). Consistently, the protein levels of MHC-I and MHC-I SIINFEKL complex induced by 4-OI and thimerosal treatment were also decreased in TFEB-knockdown EG7 cells (Fig. 3F; Appendix Fig. S3J). Importantly, the LacZ activity and IFNγ production was markedly attenuated when T cells were co-cultured with TFEB-knockdown EG7 cells as compared with that of control EG7 cells after 4-OI or thimerosal treatment (Fig. 3G–I; Appendix Fig. S3K). Conversely, overexpression of TFEB in EG7 cells increases the *B2m* expression after 4-OI or thimerosal treatment (Appendix Fig. S3L). The LacZ activity was markedly increased when T cells were co-cultured with TFEB-overexpression EG7 cells as compared with that of control EG7 cells after 4-OI or thimerosal treatment (Appendix Fig. S3M). Taken together, these results suggested that the itaconate induced tumor immunogenicity by upregulating antigen presentation via TFEB.

## RIPK3-IRF1 axis mediates thimerosal-induced itaconate production

Previous reports show that classical inflammatory pathways mediate IRG1 induction by bacterial infection and LPS in macrophages (Michelucci et al, 2013; Shi et al, 2005). We next explored how thimerosal induces IRG1 expression in tumor cells. Firstly, we analyzed the RNA-seq data from thimerosal-treated EG7 cells via Gene Set Enrichment Analysis (GSEA) to identify the key transcription factors induced by thimerosal. Thimerosal treatment enriched multiple transcription factors (Appendix Fig. S4A), among which IRF1 was recently shown as a key factor mediating IRG1 expression in neurons to clear Zika infection (Daniels et al, 2019). Thus, we tested whether IRF1 participates in thimerosal-induced IRG1 expression. We verified that the IRF1 transcription function was upregulated by thimerosal treatment (Fig. 4A). Moreover, thimerosal treatment significantly increased nuclear translocation of IRF1 in EG7 and TC-1 cells, indicating its transcriptional activation (Fig. 4B; Appendix Fig. S4B). Consistently, the ChIP-qPCR result showed that the thimerosal treatment also enhanced IRF1 binding to the IRG1 gene promoter region (Fig. 4C). Furthermore, we knocked out IRF1 and found thimerosal-induced IRG1 expression in mRNA and protein levels were both significantly attenuated in IRF1 KO cells (Fig. 4D–F). Consequently, the itaconate production induced by thimerosal in IRF1 KO cells was also attenuated (Fig. 4G). Importantly, the LacZ activity, IFNγ production, CD25, and intracellular GZMB expression on T cells were also markedly attenuated when cells were co-cultured with thimerosal-treated IRF1 KO EG7 cells as compared with WT EG7 cells (Fig. 4H; Appendix Fig. S4C,D). Supplementation of 4-OI in IRF1 KO cells rescued thimerosal-induced tumor immunogenicity (Appendix Fig. S4E), indicating that defective itaconate production is responsible for the impaired immunogenicity of IRF1 KO cells.

Previous studies suggest NF-κB, type I interferon signaling and RIPK3 could induce IRF1 activation (Daniels et al, 2019; Feng et al, 2021). We thus tested which upstream signaling mediated thimerosal-induced IRF1 activation. Inhibition of NF-κB and type I interferon signaling via chemical inhibitor or antibody blockade did not impair thimerosal-induced tumor immunogenicity (Appendix Fig. S5A,B). We next investigated whether RIPK3

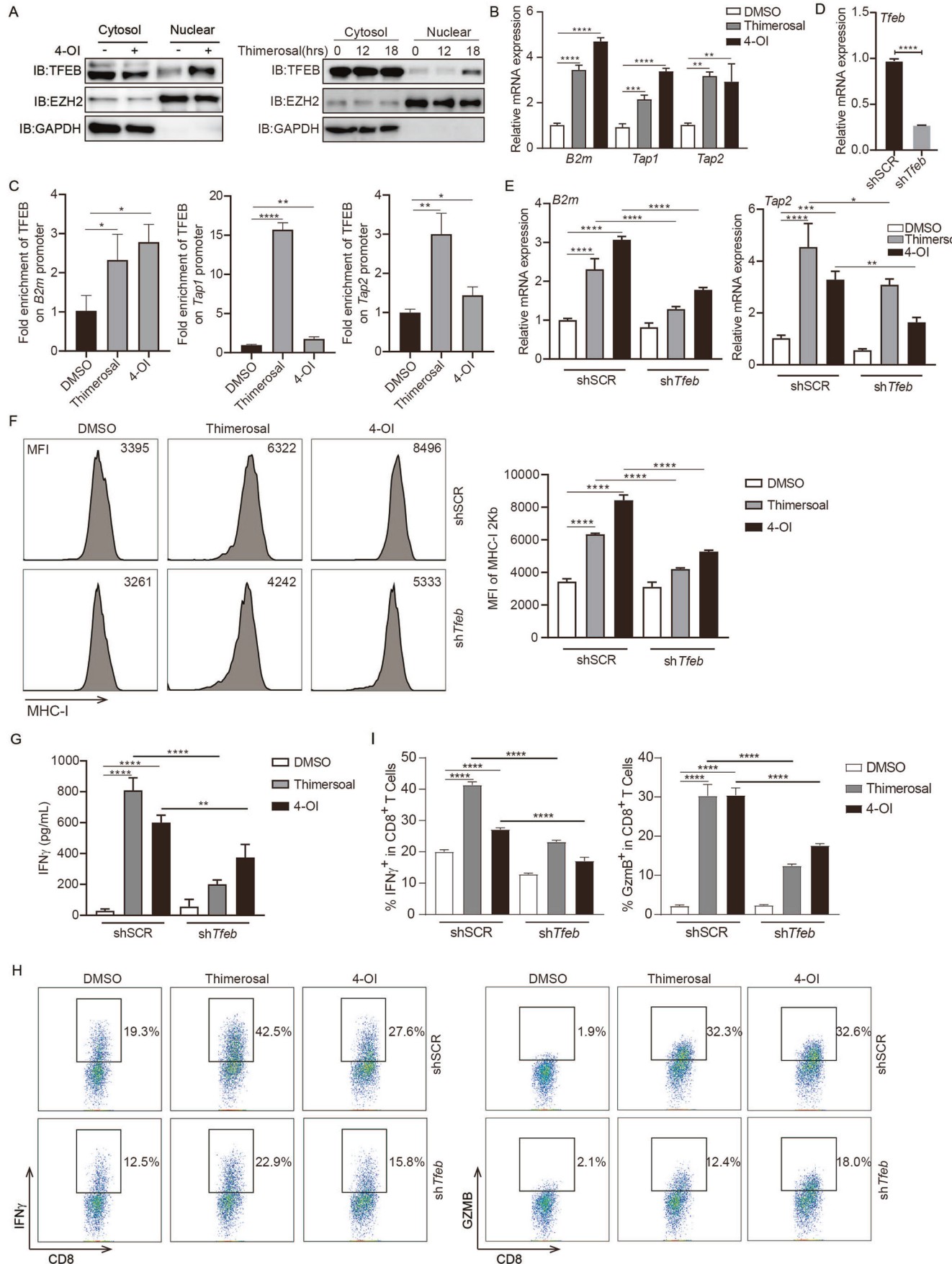

**Figure 3. Itaconate-induced tumor immunogenicity by upregulating TFEB-directed antigen presentation.**

(A) EG7 cells were treated with 4-OI (500 μM, 8 h) or thimerosal (10 μM) for the indicated time, and then the expression of TFEB in cytosol and nuclear was detected by WB. (B) EG7 cells were treated with thimerosal (10 μM) or 4-OI (500 μM) for 18 h, then the antigen presentation genes expression levels were measured by qPCR. ****$P < 0.0001$($B2m$), ***$P = 0.0002$, ****$P < 0.0001$($Tap1$), **$P = 0.004$, **$P = 0.0074$ ($Tap2$), in one-way ANOVA analysis of variance with Bonferroni's post-test. (C) EG7 cells were treated with 4-OI (500 μM, 8 h) or thimerosal (10 μM, 18 h), and then the TFEB binding on the promoter of $B2m$, $Tap1$, and $Tap2$ was measured by ChIP-qPCR. *$P = 0.0485$, *$P = 0.0137$($B2m$), ****$P < 0.0001$, **$P = 0.005$ ($Tap1$), **$P = 0.0031$, *$P = 0.031$ ($Tap2$), in Student's $t$ test. (D, E) EG7 cells expressing shSCR or sh$Tfeb$ were treated as (C) described, then the antigen presentation genes expression level was measured by qPCR. ****$P < 0.0001$ (D), in Student's $t$ test. ****$P < 0.0001$, ($B2m$), ****$P < 0.0001$, ****$P < 0.0001$, ***$P = 0.0003$, **$P = 0.0061$, *$P = 0.0153$, ($Tap2$), in one-way ANOVA analysis of variance with Bonferroni's post-test. (F) EG7 cells expressing shSCR or sh$Tfeb$ were treated as (C) described, then the expression levels of MHC-I were detected by FACS. ****$P < 0.0001$, in one-way ANOVA analysis of variance with Bonferroni's post-test. (G–I) EG7 cells expressing shSCR or sh$Tfeb$ were treated as (C) described, then followed by co-culture with OT-I for another 24 h, then the supernatant IFNγ levels were measured, and the intracellular expression levels of GZMB and IFNγ in OT-I cells were measured by FACS. ****$P < 0.0001$, **$P = 0.0052$ (G), ****$P < 0.0001$, (I, left panel), ****$P < 0.0001$, (I, right panel), in one-way ANOVA analysis of variance with Bonferroni's post-test. The graph is shown as mean ± SD of $n = 3$ for all panels. Source data are available online for this figure.

participated in thimerosal-induced IRF1 transcription. We first tested whether thimerosal could induce RIPK3 activation. We set up a RIPK3 dimerization reporter assay to reflect RIPK3 oligomerization and activation (Appendix Fig. S6A–C). Of note, among the drugs that induced tumor cell death or growth inhibition in our previous study, only thimerosal induced the RIPK3 dimerization reporter activation (Appendix Fig. S6D). Next, to confirm whether RIPK3 activation is required for thimerosal-induced IRF1 transcription, we generated RIPK3 KO cell line using the CRISPR/Cas9 technique and confirmed that RIPK3 protein expression was markedly reduced in RIPK3 KO EG7 cells (Fig. 4I). Strikingly, RIPK3 KO abolished the nuclear enrichment of IRF1 induced by thimerosal (Fig. 4J). Moreover, thimerosal-induced IRG1 upregulation and itaconate production were significantly decreased in RIPK3 KO cells (Fig. 4K–M). Consistently, Thimerosal treatment increased the surface expression of MHC-I–SIINFEKL complex on WT EG7 cells, but not on RIPK3 KO cells (Appendix Fig. S6E). LacZ activity, supernatant IFNγ production, intracellular GZMB and IFNγ expression, and T cell proliferation of B3Z or OT-I T cells were all markedly attenuated when co-cultured with thimerosal-treated RIPK3 KO EG7 cells (Fig. 4N; Appendix Fig. S6F–H). Taken together, these results demonstrated that the RIPK3-IRF1 axis mediates thimerosal-induced IRG1 expression, thereby inducing itaconate production to enhance tumor immunogenicity.

## ROS mediates thimerosal-induced RIPK3-IRF1 signaling for itaconate production

Intracellular ROS may activate RIPK3 in tumor cells (Wang et al, 2012), and thimerosal is known to induce ROS in tumor cells (Ozturk et al, 2022). In line with this, the Gene Ontology (GO) results of RNA-seq data also identified the ROS metabolic process and associated genes were significantly upregulated in EG7 cells after the thimerosal treatment (Fig. 5A,B; Appendix Fig. S7A,B). We then verified that thimerosal treatment indeed led to a significant increase in cellular ROS (Fig. 5C), ROS pathway-associated gene expression (Appendix Fig. S7C), and nuclear translocation of NRF2, a key ROS-responsive transcription factor (Appendix Fig. S7D). Consistently, co-treatment of a ROS scavenger, N-acetyl-L-cysteine (NAC), not only completely abolished thimerosal-enhanced cellular ROS (Appendix Fig. S7E), but also ROS-associated gene expression (Appendix Fig. S7F) and NRF2 nuclear localization (Appendix Fig. S7G).

Moreover, NAC almost completely blocked thimerosal-induced RIPK3-BiLC reporter activation and attenuated thimerosal-enhanced IRF1 nuclear translocation (Fig. 5D,E; Appendix Fig. S7H). Notably, IRG1 expression and itaconate production induced by thimerosal were almost abolished by NAC (Fig. 5F–H). NAC also significantly blocked thimerosal-induced tumor immunogenicity, as the co-cultured B3Z or OT-I cells showed lower levels of activation and proliferation in the NAC and thimerosal combination treatment group (Fig. 5I–L; Appendix Fig. S7I). Taken together, these data showed that ROS played an essential role in thimerosal-induced RIPK3-IRF1 axis activation to induce itaconate production, thereby triggering tumor immunogenicity.

## Tumor-intrinsic itaconate production induced by thimerosal inhibits tumor growth and activates tumor immune microenvironment

Our previous data suggested that thimerosal was a specific itaconate inducer in tumor cells but not macrophages. We then examined the impact of this itaconate inducer treatment on tumor growth in vivo. As thimerosal is a mercury compound, that poses toxic concerns in clinical application (Hurley et al, 2010), we chose the intratumor injection strategy to avoid its potential side effects and expect it would act on tumor cells directly. Intratumor administration of thimerosal significantly inhibited tumor growth of the EG7 model (Fig. 6A,B; Appendix Fig. S8A). As expected, thimerosal induced IRG1 expression in the tumor setting but not in the infiltrated immune cells (Appendix Fig. S8B). Strikingly, thimerosal had no inhibitory effect on the tumor growth of EG7 tumors established on T cell-deficient nude mice, suggesting an essential role of T cells in thimerosal-induced tumor control (Fig. 6C; Appendix Fig. S8C). In line with the results on nude mice, pretreatment with a CD8 depletion antibody abolished thimerosal-induced EG7 tumor inhibition on B6 mice, further confirming the essential role of CD8[+] T cells in thimerosal-mediated anti-tumor effect (Fig. 6D; Appendix Fig. S8D). Flow cytometry analysis found that thimerosal treatment increased the CD8[+] T cells infiltrating tumor tissues (Fig. 6E). A higher proportion of tumor-infiltrating CD8[+] T cells in the thimerosal treatment group expressed the T cell activation marker CD69, CD25, and effector molecules GZMB and IFN-γ, as compared with the control group; Meanwhile, thimerosal treatment decreased the infiltrated frequency of macrophages but not that

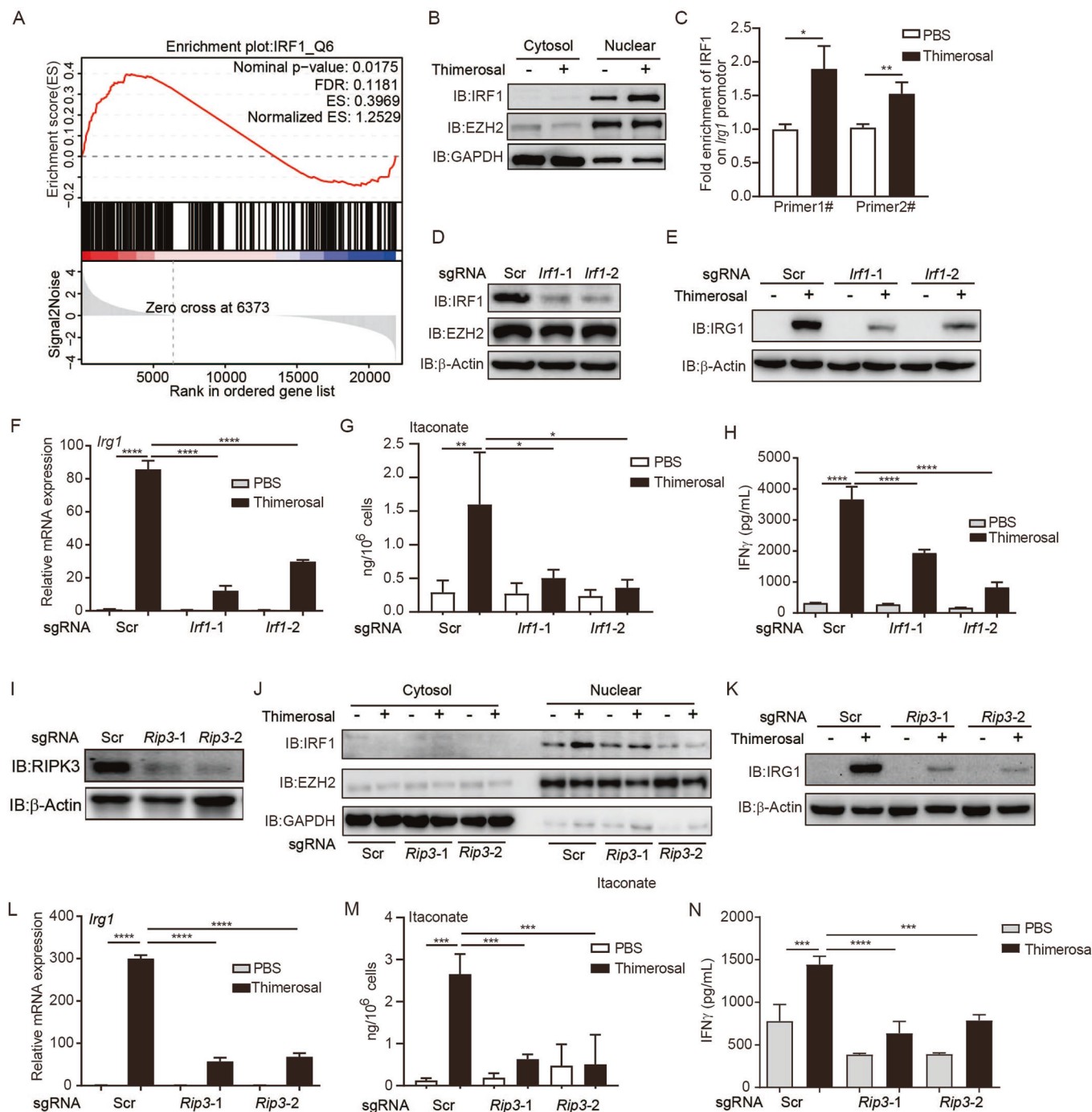

of CD4+ T cells, NK cells or B cells (Fig. 6F–J; Appendix Fig. S8E–L). Moreover, IRG1 deficiency in EG7 tumor cells significantly impaired tumor response to thimerosal treatment (Fig. 6K–L; Appendix Fig. S8M). KO IRG1 not only abolished the thimerosal inducing MHC-I or antigen presentation in tumor setting but also T cell infiltrating and activation in tumor tissue (Fig. 6M–P; Appendix Fig. S7N–P). Taken together, these results support our preclinical findings that tumor-intrinsic IRG1 expression enhances tumor immunogenicity and promotes an immune-activating tumor microenvironment.

## Tumor-intrinsic itaconate induction by thimerosal sensitizes tumor response to immunotherapy

We next examined the therapy efficacy of this itaconate inducer in combination with immunotherapy, including adoptive T cell therapy or anti-PD1 therapy. To test the combination effect, mice with established subcutaneous EG7 tumors were treated with thimerosal in combination with OVA-specific OT-I T cells or anti-PD1 antibody. Adoptive OT-I T cell transfer alone or thimerosal alone partially inhibited tumor growth, but anti-PD1 alone didn't

**Figure 4. RIPK3-IRF1 axis mediates thimerosal-induced itaconate production.**

(A) GSEA analysis shows thimerosal enhances IRF1 transcription. (B) EG7 cells were treated with thimerosal (10 μM) for 8 h, and then the protein level of IRF1 in nuclear and cytosol was detected by WB. (C) The IRF1 binding on the IRG1 promoter was measured by ChIP-qPCR after the cells were treated with thimerosal (10 μM) or PBS. *$P = 0.011$, **$P = 0.0078$, in Student's $t$ test. (D–F) EG7 sgIrf1 cells were treated with thimerosal (10 μM) for 16 h, and then the expression level of IRG1 was measured by WB or qPCR. ****$P < 0.0001$ in one-way ANOVA analysis of variance with Bonferroni's post-test. (G) EG7 sgIrf1 cells were treated with thimerosal (10μM) for 20 h, and then the production of itaconate was measured by HPLC. **$P = 0.0076$, *$P = 0.0292$, *$P = 0.0117$, in one-way ANOVA analysis of variance with Bonferroni's post-test. (H) EG7 sgIrf1 cells were treated with thimerosal (10 μM) for 16 h, followed by co-culture with OT-I for another 24 h, and then the supernatant IFNγ was measured by ELISA. ****$P < 0.0001$, in one-way ANOVA analysis of variance with Bonferroni's post-test. (I, J) EG7 sgRipk3 cells were treated with thimerosal (10 μM) for 8 h, and then the protein level of IRF1 in nuclear and cytosol was detected by WB. (K, L) EG7 sgRipk3 cells were treated with thimerosal (10 μM) for 16 h, and then the expression level of IRG1 was measured by WB and qPCR. ****$P < 0.0001$, in one-way ANOVA analysis of variance with Bonferroni's post-test. (M) EG7 sgRipk3 cells were treated with thimerosal (10 μM) for 18 h, and then the production of itaconate was measured by HPLC. ***$P = 0.0001$, ***$P = 0.0009$, ***$P = 0.0005$, in one-way ANOVA analysis of variance with Bonferroni's post-test. (N) EG7 sgRipk3 cells were treated with thimerosal (10 μM) for 16 h, followed by co-culture with OT-I for another 24 h, and then the supernatant IFNγ levels were measured by ELISA. ***$P = 0.0001$, ****$P < 0.0001$, ***$P = 0.0001$, in one-way ANOVA analysis of variance with Bonferroni's post-test. The graph is shown as mean ± SD of $n = 3$ for all panels. Source data are available online for this figure.

inhibit EG7 tumor growth. Strikingly, mice treated with thimerosal in combination with OT-I or anti-PD1 achieved significantly enhanced tumor inhibition (Fig. 6Q,R; Appendix Fig. S9A,B). Consistent with the enhanced therapeutic efficacy, the lymph node cells from those mice with complete tumor regression (treated with thimerosal in combination with OT-I) showed much higher expression levels of effector molecules including IFNγ and GZMB compared with that of lymph node cells from naïve mice when stimulated with OT-I antigen peptide (SIINFEKL) or thimerosal-treated EG7 cells (Appendix Fig. S9C,D). Taken together, these data suggested that inducing tumor-intrinsic IRG1 expression and itaconate production by thimerosal is a potentially powerful approach to improving cancer immunotherapy efficacy.

## Discussion

Itaconate plays a plethora of roles in macrophages, such as antimicrobial, anti-inflammatory, and promotes tumor progression (Chen et al, 2020; Chen et al, 2023; Gu et al, 2023; Mills et al, 2018; Weiss et al, 2018; Zhao et al, 2022). However, its production and effect on tumor cells remain largely unknown. In this study, we demonstrated that increased itaconate levels could trigger tumor immunogenicity, contrasting its typically immune-suppressive role in macrophages. The induction of tumor-intrinsic itaconate production through thimerosal was found to enhance the anti-tumor immune response. These results highlight a novel function of tumor-intrinsic IRG1/itaconate in promoting tumor immunogenicity.

Interestingly, we identified a tumor-specific itaconate inducer, thimerosal, which only induces IRG1 expression in EG7 tumor cells but not macrophage. Itaconate could be induced by various signaling pathways in macrophages (Chen et al, 2020; Michelucci et al, 2013), including the LPS-activated TLR4-NF-κB axis, the STING/MyD88/IRG1 axis, and by the microRNA miR-210 deficiency (Chen et al, 2022; Michelucci et al, 2013; Virga et al, 2021). Additionally, a recent study has demonstrated that TFEB can also stimulate IRG1 expression and itaconate production during bacterial infection in macrophages (Schuster et al, 2022). Our results suggest thimerosal-induced itaconate production in EG7 cells occurs through the ROS-RIPK3-IRF1 axis. RIPK3 serve as a crucial mediator in the necroptosis pathway (He et al, 2009), impacting liver cancer progression and chemotherapy-induced immunogenic cell death, with its expression levels correlating with

immune infiltration in breast cancer tissues (Stoll et al, 2017; Yang et al, 2016). Our results showed RIPK3, rather than NF-κB or type-I interferon signaling, acted as the upstream adaptor for IRF1-mediated IRG1 expression. A recent report indicates that ZBP1 acts as a sensor for the Zika virus, leading to the activation of RIPK3 and the promotion of an antiviral metabolic state dependent on IRF1 in neuron cells (Daniels et al, 2019). Our data also demonstrated that thimerosal only indued IRG1 expression in EG7 cells but not in macrophages, possibly due to varying induction mechanisms in different cell types. For instance, macrophages primarily rely on upstream NF-κB signaling for LPS-induced IRG1 expression, while the RIPK3-IRF1 axis may serves as the primary pathway for itaconate production in non-macrophage cells such as tumor cells or neurons. Another potential explanation is that the elevated baseline levels of reactive oxygen species (ROS) in tumor cells may render these cells more susceptible to ROS-inducing agents like thimerosal, leading to the activation of downstream signaling pathways, including RIPK3 (Trachootham et al, 2009; Trachootham et al, 2006; Zhou et al, 2003). Our future research aims to investigate additional tumor cell lines with intact signaling pathways and utilize these cell lines to confirm the role of the IRG1-itaconate axis in promoting tumor immunogenicity both in vitro and in vivo. Additionally, we observed that BMDM cells were more prone to thimerosal-induced cell death than tumor cells. Notably, the level of LPS-induced itaconate in macrophages exceeds that of tumor cells induced by thimerosal in our study, suggesting that varying levels of itaconate induced by different upstream signaling pathways may result in divergent biological functions across different cell types. Our results indicate that the production of itaconate in tumors stimulated by thimerosal is sufficient to elicit tumor immunogenicity, yet may not elicit immune suppression within the tumor microenvironment due to its limited secretion into the extracellular space from tumor cells, possibly attributed to the low expression of the itaconate exporter ABCG2 (Chen et al, 2024) (Appendix Fig. S10A).

Previous studies also reported that itaconate can directly inhibit bacteria isocitrate lyase for antimicrobial purposes, as well as interact with RAB32 in macrophages to produce itaconate in vacuole for defense against Salmonella (Chen et al, 2020). Additionally, itaconate has been shown to target TFEB to increase lysosome biogenesis for bacteria removal in macrophages (Zhang et al, 2022). Furthermore, it modulates the oxoglutarate receptor 1

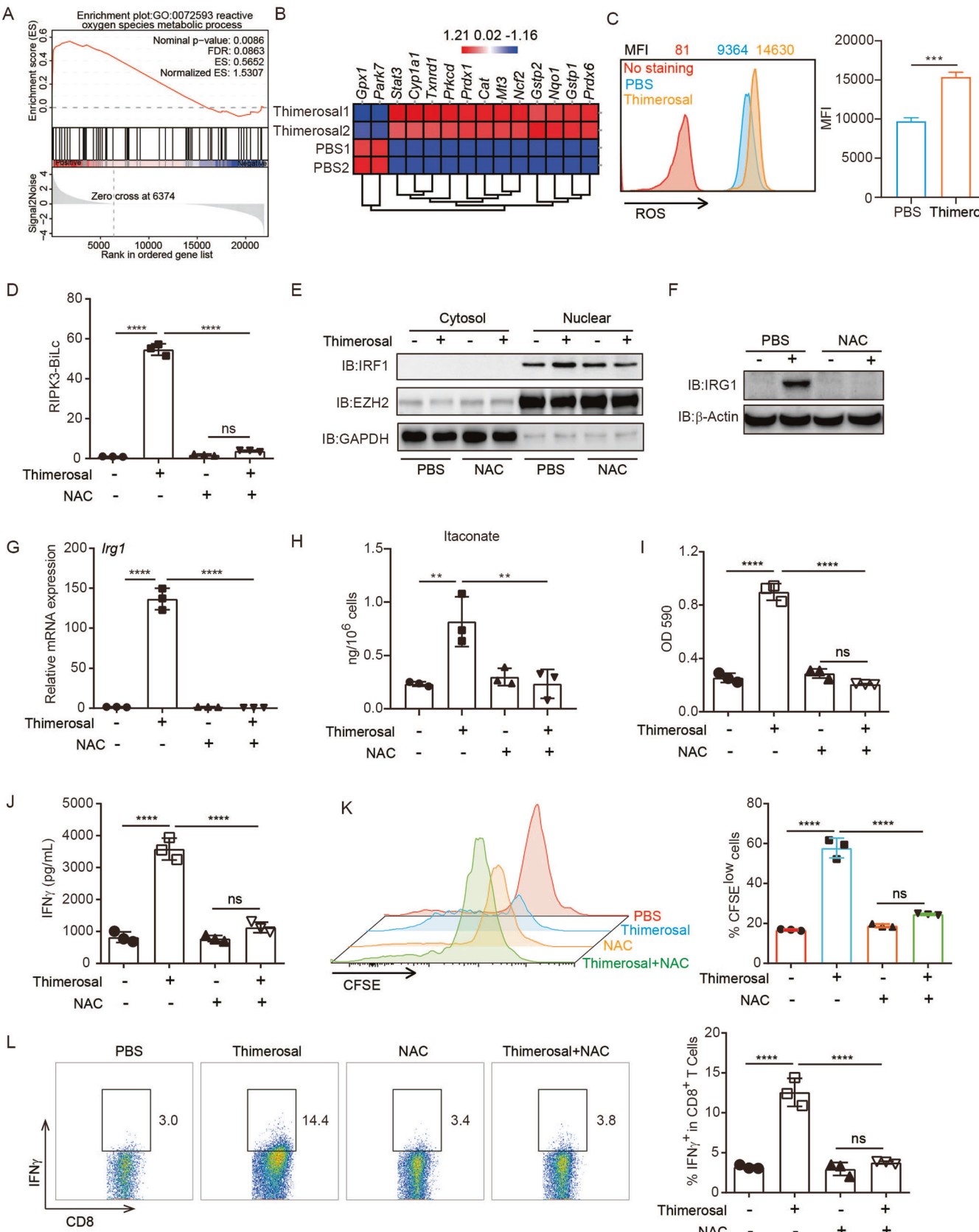

Figure 5.   ROS plays an essential role in thimerosal-induced itaconate production.

(A) Enrichment plots of ROS pathway in EG7 cells with thimerosal treatment. (B) Heat map of gene expression of ROS pathways in EG7 cells with thimerosal treatment. (C) EG7 cells were treated with thimerosal (10 µM) for 8 h, and then the intracellular ROS were detected by DCFDA staining and FACS analysis. ***$P = 0.0002$, in Student's *t* test. (D) HT29 RIPK3-BiLC reporter cells were treated with thimerosal (10 µM) alone or thimerosal plus NAC (1 mM) for 8 h, then the luciferase activity was measured. ****$P < 0.0001$, in one-way ANOVA analysis of variance with Bonferroni's post-test. (E) EG7 cells were treated with thimerosal (10 µM) or thimerosal plus NAC (1 mM) for 8 h, then the protein level of IRF1 in nuclear and cytosol was detected by WB. (F, G) EG7 cells were treated as (E) for 18 h, and then the expression level of IRG1 was measured by WB and qPCR. ****$P < 0.0001$, in one-way ANOVA analysis of variance with Bonferroni's post-test. (H) EG7 cells were treated as (E) for 18 h; the production of itaconate was measured by HPLC. **$P = 0.0058$, **$P = 0.0058$, in one-way ANOVA analysis of variance with Bonferroni's post-test. (I–L) EG7 cells were treated as (E) for 16 h, followed by co-culture with B3Z or OT-I cells for an additional 24 h, then the LacZ activity or supernatant IFNγ levels were measured. T cell proliferation and intracellular expression levels of IFNγ in OT-I cells were measured by FACS. ****$P < 0.0001$ (I–L), in one-way ANOVA analysis of variance with Bonferroni's post-test. The graph is shown as mean ± SD of $n = 3$ for all panels. Source data are available online for this figure.

(OXGR1) protein to aid in clearing bacteria through the promotion of mucociliary clearance (Zeng et al, 2023). Our study demonstrated that itaconate upregulates TFEB-mediated antigen presentation to enhance tumor immunogenicity. Interestingly, an early study showed that TFEB-induced lysosome biogenesis attenuates MHC-I but enhances MHC-II antigen presentation in DCs (Samie and Cresswell, 2015). Conversely, a more recent study found that upregulation of TFEB in Merkel cell carcinoma led to increased expression of HLA-ABC in tumor cells (Lee et al, 2022). Our results revealed that the knockdown of TFEB resulted in decreased expression of MHC-I and the MHC-I/SIINFEKL complex on EG7 tumor cells following stimulation with itaconate and thimerosal. These findings suggest that TFEB may have distinct roles in modulating antigen presentation in tumor cells compared to DCs, or itaconated-stimulated TFEB's nuclear function preferentially induce MHC-I antigen presentation molecule transcription. Future investigation is warranted to delineate the tumor-specific antigen processing and immunogenicity regulation by itaconate.

Accumulating evidence suggests that immune "hot" tumor microenvironment may be indicative of improved efficacy of immunotherapy (Bruni et al, 2020; Galluzzi et al, 2018). Our in vivo experiments demonstrated that increasing intratumoral levels of itaconate enhances infiltration and activation of immune cells, while IRG1-deficient EG7 tumors exhibited diminished responses to thimerosal treatment. Moreover, our study illustrated that the addition of thimerosal to OT-I or anti-PD1 led to increased effectiveness against tumors in a murine model. To validate our results from murine models, we sought to investigate the expression of IRG1 and the immune microenvironment in human cancer specimens. Our analysis revealed diminished levels of IRG1 expression in diverse human tumor tissues and tumor cells (Appendix Fig. S10B,C). We hypothesized that the signaling pathways responsible for inducing IRG1 expression may be impaired or altered in these tumor cells. On the other hand, tumor cells exhibiting high levels of IRG1 expression may be correlated with heightened tumor immunogenicity, potentially leading to their eradication by immunosurveillance during tumorigenesis. Conversely, tumor cells displaying diminished IRG1 expression may be more prone to evade immune detection. In contrast to our findings, several recent reports showed that IRG1 is predominantly upregulated in MDSC cells such as neutrophils rather than tumor cells within human and mouse tumor tissue (Chen et al, 2023; Zhao et al, 2022; Zhao et al, 2023). Additionally, itaconate released by MDSCs has been shown to impede the activity of CD8 + T cells and exert immunosuppressive effects. Nevertheless, these findings do not preclude the possibility of a tumor-intrinsic

function of itaconate in influencing tumor immunogenicity. It is a common observation that the same metabolite can have contrasting effects on tumor cells as opposed to immune cells. For instance, the uptake of succinate by T cells has been shown to inhibit their effector function (Gudgeon et al, 2022), while a recent study demonstrated that inhibiting mitochondrial complex II could lead to succinate accumulation within tumor cells, subsequently modulating histone lysine methylation on antigen-presenting genes to enhance tumor cell immunogenicity (Mangalhara et al, 2023). Therefore, the intracellular metabolite may exhibit distinct effects on tumor cells compared to its extracellular impact on adjacent immune cells. Our inability to detect extracellular itaconate in the culture media of tumor cells following thimerosal treatment suggests that tumor-derived itaconate induced by thimerosal is likely retained within the cells to augment tumor immunogenicity, rather than being released into the tumor microenvironment and promote immune suppression.

Thimerosal has been utilized extensively as a vaccine preservative to inhibit microbial growth (Golos and Lutynska, 2015). Additionally, thimerosal has been shown to trigger tumor cell apoptosis through activation of mitochondrial or p38 MAPK signaling pathways (Liu et al, 2007), while concurrently augmenting reactive oxygen species (ROS) levels by depleting cellular glutathione (Ozturk et al, 2022). We also noticed tumor cell death induced by long-term thimerosal treatment. Importantly, we demonstrated that thimerosal can elevate intracellular ROS levels, thereby activating the RIPK3/IRF1/IRG1 axis to enhance tumor immunogenicity. Consistent with our findings, recent studies have similarly demonstrated the necessity of reactive oxygen species (ROS) induction for enhancing the immunogenicity of nanoparticle-formulated tumor vaccines (Liang et al, 2018; Wang et al, 2016a). While thimerosal has historically been utilized as a vaccine preservative rather than for oncological purposes, clinical data about its efficacy in cancer treatment remains elusive. Based on our investigations, we posit that thimerosal exhibits significant potential for advancing cancer therapeutics either independently or in conjunction with existing immunotherapeutic approaches.

Overall, our findings demonstrated a novel function of tumor-intrinsic IRG1/itaconate in enhancing tumor immunogenicity and anti-tumor immunity. We identified thimerosal as a tumor-intrinsic itaconate inducer that can enhance tumor immunogenicity and sensitize tumors to immunotherapy. These findings highlight the specific role of IRG1/itaconate in promoting tumor immunogenicity and suggest that targeting the tumor-intrinsic IRG1-itaconate axis could be a viable strategy for improving cancer immunotherapy. Additionally, our study offers a practical approach for enhancing immune response in cancer treatment.

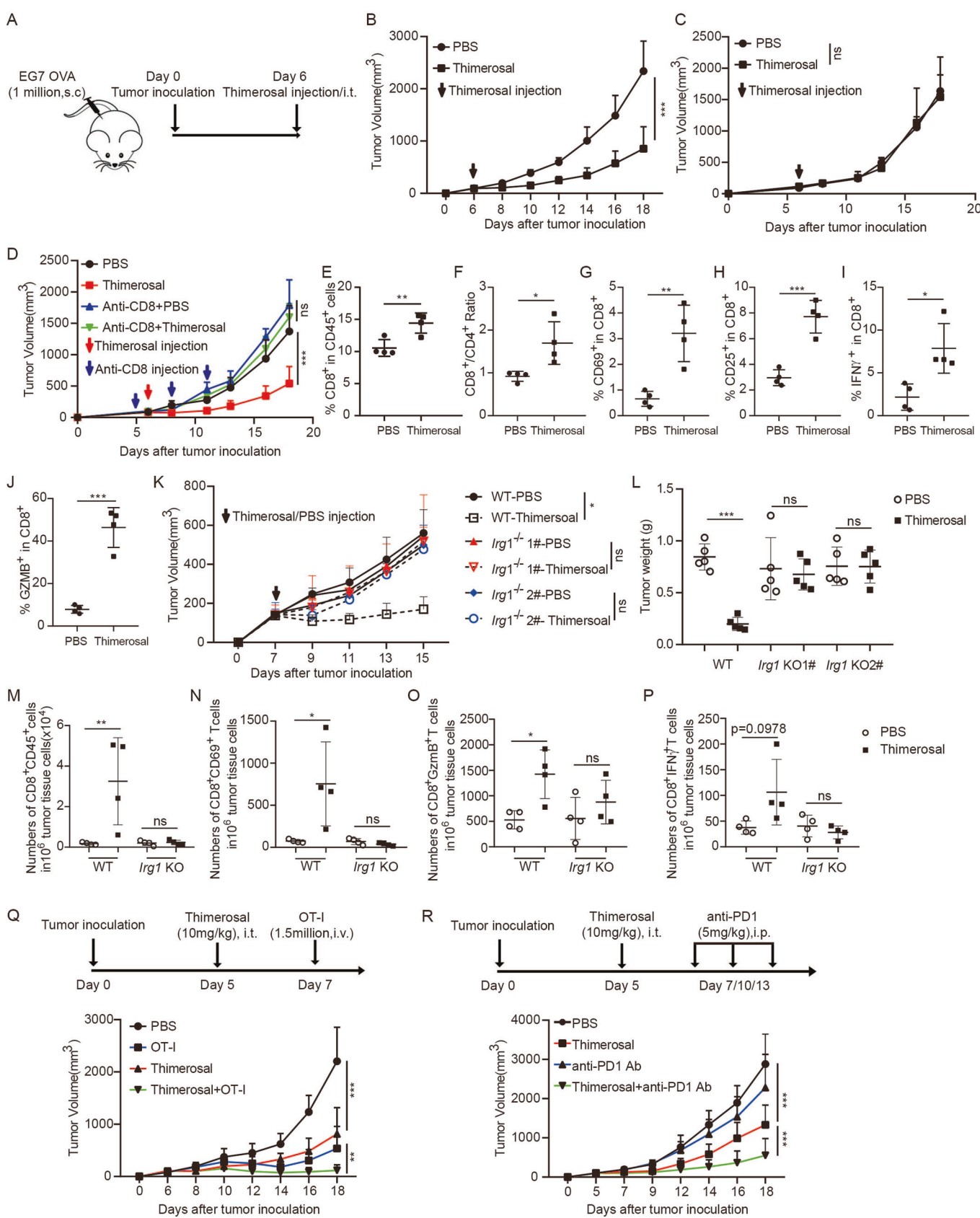

**Figure 6. IRG1/itaconate induction in tumor cells by thimerosal inhibits tumor growth, activates tumor immune environment, and sensitizes tumor response to immunotherapy.**

(A) A schematic illustration of thimerosal treatment regimen on EG7 tumor model. (B) EG7 cells (1 million per mouse) were subcutaneously (s.c.) inoculated in the flank of B6 mice, followed by intratumoral injection of thimerosal or PBS on day 6, and then the tumor growth was recorded, $n = 6$ per group. ***$P < 0.0001$, in two-way ANOVA analysis of variance with Bonferroni's post-test. (C) EG7 cells (1 million per mouse) were s.c. inoculated in the flank of nude mice, followed by intratumoral injection of thimerosal or PBS on day 6, then the tumor growth was recorded, $n = 5$ per group. $P > 0.9999$, in two-way ANOVA analysis of variance with Bonferroni's post-test. (D) EG7 tumors were treated as in (B), together with intraperitoneal (i.p.) injection of anti-CD8 depletion antibody on day 5, 8, and 11, then the tumor growth was recorded, $n = 5$ per group. ***$P < 0.0001$, in two-way ANOVA analysis of variance with Bonferroni's post-test. (E–J) EG7 tumors were treated as in (B), and the tumors were harvested on day 10 for immunotyping analysis by FACS. $n = 4$ per group. **$P = 0.0088$ (E), *$P = 0.0223$ (F), **$P = 0.0042$ (G), ***$P = 0.0005$ (H), *$P = 0.0132$ (I), ***$P = 0.0002$ (J), in Student's $t$ test. (K, L) WT or two lines of $Irg1^{-/-}$ EG7 cells ($Irg1^{-/-}$1#, $Irg1^{-/-}$2#) were subcutaneously (s.c., 1 million cells per mouse) inoculated in the flank of B6 mice, followed by intratumoral injection of thimerosal or PBS on day 7, then the tumor growth was recorded. Tumor weights were recorded on day 15. $n = 5$ per group. *$P = 0.0245$, in two-way ANOVA analysis of variance with Bonferroni's post-test (K). ***$P = 0.0001$, in one-way ANOVA analysis of variance with Bonferroni's post-test (L). (M–P) WT or $Irg1^{-/-}$ EG7 cells were treated as in (K), and the tumors were harvested on day 11 for immunotyping analysis by FACS. $n = 4$ per group. **$P = 0.0099$ (M), *$P = 0.0142$ (N), *$P = 0.0428$ (O), $P = 0.0978$ (P), in one-way ANOVA analysis of variance with Bonferroni's post-test (L). (Q) B6 mice bearing subcutaneous EG7 tumors were treated with thimerosal alone or in combination with OT-I; tumor growth was recorded, $n = 5$ for each group. ***$P = 0.0001$, **$P = 0.0061$ in two-way ANOVA analysis of variance with Bonferroni's post-test. (R) B6 mice were s.c. inoculated with EG7 cells (1 million), then were treated with thimerosal alone or in combination with anti-PD1; tumor growth was recorded, $n = 5$ for PBS and anti-PD1 groups, $n = 6$ for the thimerosal group and $n = 7$ for the combination group. ***$P < 0.001$, in two-way ANOVA analysis of variance with Bonferroni's post-test. Tumor volume and the graph is shown as mean ± SD. Source data are available online for this figure.

# Methods

## Reagents and tools table

| Reagent/resource | Reference or source | Identifier or catalog number |
|---|---|---|
| RPMI 1640 | Invitrogen | C11875500BT |
| DMEM | Invitrogen | C11995500BT |
| FBS | TransGen Biotech | FS401-02 |
| Thimerosal | Selleck | S3646 |
| NAC | Selleck | S1623 |
| Human TNFα | Peprotech | 315-01A |
| Birinapant | Apexbio Inc | A4219 |
| LCL161 | Apexbio Inc | A3541 |
| z-VAD-FMK | Apexbio Inc | A1902 |
| Necrosulfonmide | Apexbio Inc | B7731 |
| Nerostain-1 | Apexbio Inc | A4213 |
| 4-Octyl itaconate | Topscience Inc | T4580 |
| Itaconic acid | Topscience Inc | T4843 |
| Digitonin | Topscience Inc | T2721 |
| Dimethyl itaconate | Sigma | 592498 |
| H2DCFDA | Invitrogen | D399 |
| Chlorophenol red β-D-galactopyranoside | Merck | 59767 |
| 16% Formaldehyde solution(v/v) | Invitrogen | 28906 |
| BSA | Sigma | A9418 |
| TRIzol | Invitrogen | 15596018 |
| HiScript III RT SuperMix for qPCR | Vazyme | R323-01 |
| SYBR Premix Kit | MIKX | MK900-10 |
| Mouse IL-2 Uncoated ELISA Kit | Invitrogen | 88-7024-88 |
| Mouse IFN gamma Uncoated ELISA Kit | Invitrogen | 88-7314-88 |
| *Renilla* luciferase assay system | Promega | E2820 |
| **Experimental models** | | |
| C57BL/6 | Vital River Laboratory | C57BL/6NCrl |

| Reagent/resource | Reference or source | Identifier or catalog number |
|---|---|---|
| BALB/c Nude | Vital River Laboratory | CAnN.Cg-Foxn1$^{nu}$/Crl |
| OT-I | Jackson Laboratory | 003831 |
| 293T | ATCC | CRL-3216 |
| EG7 | Gifted by Dr. Haidong Tang (Tsinghua University) | Methods |
| B3Z | Gifted by Dr. Nilabh Shastri (Johns Hopkins University) | Methods |
| E0771 | Gifted by Dr. Lin Tian (Sun Yat-Sen University Cancer Center) | Methods |
| MC38 | Gifted by Dr. Xuanming Yang (Shanghai Jiaotong University) | Methods |
| 4T1 | ATCC | CRL-2539 |
| CT26 | ATCC | CRL-2638 |
| T24 | Gifted by Dr. Jing Tan (Sun Yat-Sen University Cancer Center) | Methods |
| J82 | Gifted by Dr. Jing Tan (Sun Yat-Sen University Cancer Center) | Methods |
| **Recombinant DNA** | | |
| RIPK3-GlucN | This study | Methods |
| RIPK3-GlucC | This study | Methods |
| TFEB-WT/TFEB C270S | Gifted by Dr. Xinjian Li (Chinese Academy of Sciences) | Methods |
| **Antibodies** | | |
| *Anti-mouse PD1* | BioXcell | *BE0146* |
| *Anti-mouse CD8* | BioXcell | *BE0004-1* |
| *CD8* | eBioscience | *25-0081-82* |
| *CD69* | Biolegend | *104514* |
| *CD25* | eBioscience | *88-7314-22* |
| *IFNγ* | eBioscience | *25-7311-82* |
| *Granzyme B (Gzm B)* | eBioscience | *48-8898-82* |
| *CFSE* | eBioscience | *65-0850-84* |
| *CD45* | eBioscience | *404-0451-82* |
| *NK1.1* | eBioscience | *45-5941-82* |
| *CD11c* | eBioscience | *17-0114-82* |
| *F4/80* | eBioscience | *11-4801-82* |

| Reagent/resource | Reference or source | Identifier or catalog number |
|---|---|---|
| CD4 | eBioscience | 47-0041-82 |
| CD3 | eBioscience | 48-0031-82 |
| MHC-I H2Kb | BD Bioscience | 756513 |
| OVA257-264 (SIINFEKL) peptide bound to H-2Kb | eBioscience | 17-5743-80 |
| actin | Santa Cruz | sc8432 |
| Flag | Sigma | A8592 |
| TFEB | CST | 83010 |
| NRF2 | CST | 12721 |
| GAPDH | Santa Cruz | Sc32233 |
| IRF1 | CST | 8478 |
| IRG1 | CST | 19857 |
| Anti-mouse IgG, HRP-linked Antibody | CST | 7076 |
| Anti-mouse IgG, HRP-linked Antibody | CST | 7074 |
| **Oligonucleotides and other sequence-based reagents** | | |
| qPCR primers | This study | Methods |
| sgRNA sequence | This study | Methods |
| ChIP qPCR primer | This study | Methods |
| **Chemicals, enzymes and other reagents** | | |
| BsmBI | New England Biolabs | NEB#R0580 |
| **Software** | | |
| GraphPad Prism V5 | GraphPad | |
| FlowJo V10 | BD Bioscience | |
| Compound Discover 3.3 | ThermoFisher | |
| **Other** | | |
| TruSeq Stranded RNA HT kit 96 samples Ribo-Zero Gold | Illumina | |
| Illumina NextSeq 2000 | Illumina | |

## Mice and reagents

Six to 8-week-old female C57BL/6J (B6) and nude mice were purchased from the Vital River Laboratory (Beijing, China). OT-I mice were obtained from The Jackson Laboratory.

HEK293 cell line was obtained from the American Type Culture Collection (ATCC). EG7 (mouse lymphoma) cell line was kindly gifted by Dr. Haidong Tang (Tsinghua University, Beijing, China). E0771 cells were kindly gifted by Dr. Lin Tian (Sun Yat-Sen University Cancer Center, China). B3Z hybridoma cells were kindly gifted by Dr. Nilabh Shastri (Johns Hopkins University, USA). MC38 cells were kindly gifted by Dr. Xuanming Yang (Shanghai Jiaotong University, China). 4T1 and CT26 cells were from ATCC. HT29, T24 and J82 cells were gifted by Dr. Jing Tan (Sun Yat-Sen University Cancer Center, China). All cell lines were routinely tested as being mycoplasma-free. The cells were maintained either with Dulbecco's Modified Eagle medium (DMEM) (Invitrogen) supplemented with 10% fetal bovine serum (FBS) and 1% penicillin–streptomycin or Roswell Park Memorial Institute (RPMI) 1640 (Invitrogen) supplemented with 1% penicillin–streptomycin and 10% FBS in a humidified atmosphere at 37 °C and 5% $CO_2$.

Thimerosal (S3646) and NAC (S1623) were purchased from Selleck, Inc. TNF-α was from Peprotech (315-01A); birinapant (A4219), LCL161 (A3541), z-VAD-FMK (A1902), necrosulfonamide (B7731) and necrostatin-1 (A4213) were all purchased from Apexbio Inc. 4-Octyl itaconate (T4580), itaconate (T4843), and Digitonin (T2721) were purchased from Topscience Inc. Dimethy itaconate (592498) was purchased from Sigma. Anti-mouse PD1 antibody (BE0146) and anti-CD8 antibody (BE0004-1) were from BioXcell Inc.

## RIPK3 dimerization reporter assay

RIPK3-BiLC reporter constructs were cloned into the lentiviral vector backbone as the previous study described (Wang et al, 2016b). GlucN (17–93 aa) and GlucC (94–185 aa) fragments (GenBank no. AY015993) were inserted into the vector as a c-terminal tag. RIPK3 open reading frames were cloned into the RIPK3-BiLC reporter system as RIPK3-GlucN or RIPK3-Gluc fusion protein. RIPK3-BiLC reporter was stably transfected into tumor cells by a lentiviral-based backbone, and the stably transfected cells were treated with indicated drugs for 8 h. Using the *Renilla* luciferase assay system (Promega E2820), the cells were lysed by *Renilla* lysis buffer, and the Gluc activity of the RIPK3-BiLC reporter was measured according to the manufacturer's instructions.

## LacZ activity measurement

The procedures for LacZ activity measurement were performed as previously described (Lee et al, 2005). Briefly, after activation, B3Z cells in the wells of a cell culture plate were lysed and freeze-thawed, to which 50 μL/well of PBS containing 0.5% bovine serum albumin and 100 μL/well of substrate solution (1 mg/mL chlorophenol red β-D-galactopyranoside) dissolved in β-galactosidase buffer were added. The plate was incubated at 37 °C for 12 h to 18 h till color development reached a proper level, followed by color intensity reading at 590 nm using a microtiter plate reader.

## T cell activation and proliferation assay

Tumor cells were treated with thimerosal, 4-Octyl itaconate, or dimethyl itaconate for the indicated time, the treatment of drugs was removed by refreshing media before co-culture with B3Z or OT-I for additional time points, after which the LacZ activity in B3Z cells was detected, supernatant level of IFNγ was measured by ELISA kit (eBioscience, 88-7314-22). T cells were stained with fluorescence-labeled antibodies against CD8 (eBioscience, 25-0081-82), CD69 (Biolegend, 104514), CD25 (eBioscience, 12-0251-82), IFNγ (eBioscience, 25-7311-82), Granzyme B (Gzm B) (eBioscience, 48-8898-82). For the T cell proliferation assay, OT-I cells were labeled by carboxyfluorescein succinimidyl amino ester (CFSE) (eBioscience, 65-0850-84) before co-culture with drug-treated tumor cells, and the proliferation of OT-I cells was measured by flow cytometry after 48 h of co-culture. For MHC-I SIINFEKL detection, EG7 cells were treated with indicated drugs for about 20 h, after which the cells were stained with an antibody recognizing SIINFEKL-H2Kb complex (25-D1.16, eBioscience, 17-5743-80) and analyzed by flow cytometry.

## Real-time PCR and RNA-seq

Total RNA was isolated using TRIzol (Invitrogen, 15596018) according to the manufacturer's instructions. RNA was reverse transcribed using the HiScript III RT SuperMix for qPCR (Vazyme, R323-01). Real-time PCR was performed using the SYBR Premix Kit (MIKX, MK900-10) and analyzed using the Bio-Rad CFX96 thermal cycler. The primer sequences used for the investigated mouse genes were as follows:

*Actin*, 5′-AGAGGGAAATCGTGCGTGAC-3′, 5′-CAATAGT-GATGACCTGGCCGT-3′;

*Gclm*, 5'-GACAAAACACAGTTGGAACAGC-3′, 5′-CAGT-CAAATCTGGTGGCATC-3′;

*Nqo1*, 5′-GCTGCAGACCTGGTGATATT-3′, 5′-ACTCTCT-CAAACCAGCCTTT-3′;

*Irg1*, 5′-GGTATCATTCGGAGGAGCAAGAG-3′, 5′-ACAGTG CTGGAGGTGTTGGAAC-3′.

*Atp6v0d2*, 5′-ACGGTGATGTCACAGCAGACGT-3′, 5′-CTCT GGATAGAGCCTGCCGCA-3′

*p62*, 5′-TCACAGATCACATTGGGGTGC-3′, 5′-AGGATGGG-GACTTGGTTGC-3′;

*Cstb*, 5′-ATAGTGGCTGGCACCAACCTCT-3′, 5′-GGAAGAC AGGGTCAAAGGCTTG-3′;

*Cstd*, 5′-TAAGACCACGGAGCCAGTGTCA-3′, 5′-CCACAG GTTAGAGGAGCCAGTA-3′;

*Tfeb*, 5′-CGCCTGGAGATGACTAACAAGC-3′, 5′-GGCAACT CTTGCTTCACCACCT-3′;

*B2m*, 5′-TTCTGGTGCTTGTCTCACTGA-3′, 5′-CAGTATGT TCGGCTTCCCATTC-3′;

*Earp1*, 5′-CTCCCACTTTTAGCAGTCCCC-3′, 5′-AAAGTCA-GAGTGCTGAGGTTTG-3′;

*Tap1*, 5′-GGACTTGCCTTGTTCCGAGAG -3′, 5′-GCTGCCA-CATAACTGATAGCGA-3′;

*Tap2*, 5′-CTGGCGGACATGGCTTTACTT-3′, 5′-CTCCCA CTTTTAGCAGTCCCC-3′.

*Abcg2*, 5′-CAGTTCTCAGCAGCTCTTCGAC-3′, 5′-TCCTCC AGAGATGCCACGGATA-3′.

For RNA-Seq, the libraries were prepared using TruSeq Stranded RNA HT kit 96 samples Ribo-Zero Gold (Illumina) following the manufacturer's instructions. RNA sequencing was performed on a NovaSeq sequencer (Illumina).

## CRISPR/Cas9 knockout and shRNA knockdown

IRF1-KO, RIPK3-KO and IRG1-KO cells were constructed through the CRISPR/Cas9 system (Sanjana et al, 2014). The sgRNA sequences were designed using the Optimized CRISPR Design (http://chopchop.cbu.uib.no/). The guide sequences used were 5′-GTGGGACTTCGTGTCCGGGC-3′, 5′-GTTGCAGGGAGATGGA AGACA-3′ for RIPK3; 5′-GCCATGCCAATCACTCGAATG-3′, 5′-GCCGACACACATCGATGGCAA-3′ for IRF1; 5′-GTATAGC GTGTCGAAGCTTGG-3′, 5′-GGAAGGGCCTGTTTACGTAC-3′ for IRG1. The sgRNA was inserted into the LentiCRISPR v2 vector, which also contained the *Streptococcus pyogenes* Cas9 nuclease gene. sgRNA lentiviral vectors were co-transfected with pspax2 and pMD2.G packaging plasmids in 293T cells. The supernatants were harvested 48 h after transfection and used for infection with tumor cells, followed by selection with puromycin for another 3 days, and then the KO effect was confirmed by

western blot analysis of whole-cell protein extracts. shRNA lentiviral vectors were packaged and infected with tumor cells, followed by puromycin selection for an additional 3 days. The knockdown effect was assessed by Western blot analysis of whole-cell protein extracts.

## Global untargeted metabolomics with liquid chromatography–mass spectrometry (LC–MS) and itaconate concentration measurement

EG7 ($5 \times 10^6$) cells were treated with thimerosal for 16 h and washed with 0.9% NaCl. Samples were then extracted with LCMS grade methanol-water-chloroform at 10:3:10 (v/v/v) (230 μL per $10^6$ cells) and spin for 15 min (15,000 rcf, 4 °C). Collect the supernatant and dried into powder. For the untargeted metabolomics assay, four biological replicates were prepared for ultrahigh-performance liquid chromatography electrospray ionization mass spectrometry (UHPLC-ESI–MS) analysis (Q Exactive, ThermoFisher Scientific). For the itaconate measurement, the sample powder and standards were reconstituted in LCMS grade acetonitrile water at 4:1 (v/v), while the concentration range of standards was 2–500 ng/mL. All the samples and standards were directly injected into the LCMS system for analysis. LCMS analysis was performed on Agilent 6495A Triple Quadrupole LC/MS coupled with Agilent 1290 Infinity II HPLC system. A BEH Amide column (2.1*100 mm, 1.7 μm, Waters) was utilized for liquid chromatography separation at 35 °C. Buffer A contained 10 mM ammonium formate and 0.3% ammonia solution in water, buffer B contained 10 mM ammonium formate and 0.3% ammonia solution in acetonitrile. The separation was conducted at 0.3 mL/min for 5 min by the elution with 70% buffer B. Mass spectrometer was operated using multiple reaction monitoring mode (MRM) and electrospray ionization (ESI) sources in negative ion mode. The following transitions and conditions were used for itaconate: $129.0 \rightarrow 85.2$, CE 5; the capillary voltage, 4000 V; sheath gas temperature, 400 °C; sheath flow, 12 L/min; nebulizer, 40 psi; gas temperature, 275 °C; and gas flow, 14 L/min. The data for the untargeted metabolomic assay obtained from the above metabolomics liquid methods were imported into Compound Discover 3.3 software for data processing and statistical analysis. By extracting the mass-to-charge ratio with a response value higher than 2e6, setting the mass deviation to 5ppm, the unknown metabolic samples were identified by molecular formula prediction, mzcloud database and ChemSpider database, and the identification results were statistically analyzed on Compound Discover 3.3 software, for principal component analysis (PCA) and Partial least squares discriminant analysis (PLS—DA).

## Western blot

The procedures for protein sample preparation from cell cultures, protein quantification, Western blot, and data analyses were performed as previously described (Wang et al, 2016b). The following antibodies were used for Western blot analyses: actin (Santa Cruz, catalog sc8432), Flag (sigma, catalog A8592), TFEB (CST, catalog 83010), NRF2 (CST, catalog 12721), EZH2 (CST, catalog 5246), GAPDH (Santa Cruz, catalog sc32233), IRF1 (CST, catalog 8478), and IRG1 (CST, catalog 19857). Protein bands were visualized by chemiluminescence using an ECL detection kit (Thermo Scientific, 32106).

## ChIP-qPCR

Cells were cross-linked with 1% formaldehyde and quenched with 0.125 M glycine. The cross-linked cells were lysed (50 mM Tris-HCl [pH 8.0], 10 mM EDTA, 1% SDS) and sonicated on ice using Bioruptor (Diagenode, Belgium). The lysates were precleared and immunoprecipitated with protein G Dynabeads (Invitrogen) and incubated with indicated antibody and beads complex overnight at 4 °C. Immunoprecipitations were eluted, reverse cross-linked, and purified for ChIP-DNA. For ChIP-qPCR analysis, the ChIP-DNA was quantitated using the SYBR Premix Kit (MIKX, MK900-10) on a Bio-Rad CFX Real-Time PCR machine. The ChIP primer sequences are as following Irg1 Primer1: 5′-GTTGGTTGGTGAC AATGGGC-3′, 5′-CTGGGCATCCCACTCTTGAA-3′; Irg1 Primer2: 5′- TCTTCTCACCTATGGCCCCT-3′, 5′-CACCTCCCCA ATGACCTCAC-3′.

B2m:5′-CTGCTAGAAGCAAGGTCAGAAA-3′, 5′-TCTCAGC CCAGAAATACAAAGG-3′; Tap1:5′-GAGAGCAAGAACTGAGA GGAAG-3′, 5′-CCTGGTCTTGTGTCACAGTAAT-3′; Tap2: 5′- C CTTCAAGTGACTCATGGGTAG-3′, 5′-AGGGCACTACAAGGA GAAATG-3′.

## Tumor growth, treatment, and analysis

For the in vivo study, EG7 tumor cells ($1 \times 10^6$ cells/mouse) were subcutaneously injected into the right flank of B6 mice or nude mice. The mice were administrated with thimerosal or PBS by intratumoral injection (10 mg/kg) on the indicated day. The tumor volume was calculated using the formula $0.5 \times$ tumor length $\times$ (tumor width)$^2$, where the longer dimension was considered as the tumor length.

For immunophenotyping analysis of the tumor microenvironment, EG7 tumor cells ($1 \times 10^6$ cells/mouse) were subcutaneously injected into the right flank of B6 mice. Then the mice were treated with thimerosal or PBS by intratumoral injection (10 mg/kg) on day 6. Tumor tissues were collected and analyzed on day 10. For analysis of immune cell populations, mouse tumors were dissociated by gentleMACS (Miltenyi Biotec) and filtered through 70 μm cell strainers to generate single-cell suspensions, then stained with fluorescence-labeled antibodies against CD45 (eBioscience, 11-0451-82), CD3 (eBioscience, 48-0031-82), CD4 (eBioscience, 47-0041-82), CD8 (eBioscience, 25-0081-82), CD69 (Biolegend, 104514), CD25 (eBioscience, 12-0251-82), IFNγ (eBioscience, 25-7311-82), Granzyme B (Gzm B) (eBioscience, 48-8898-82), NK1.1 (eBioscience, 45-5941-82), CD11c (eBioscience, 17-0114-82), F4/80 (eBioscience, 11-4801-82). Fluorescence data were acquired using a BD LSR Fortessa cytometer and analyzed using the FlowJo software, V10.

For combination therapy, EG7 tumor cells ($1 \times 10^6$ cells/mouse) were subcutaneously injected into the right flank of B6 mice. Then the mice were randomly divided into four groups; thimerosal or PBS was administered by intraperitoneal injection as indicated, followed by three times intraperitoneal injection of anti-PD1 (5 mg/kg) or once of intravenous injection of OT-I cells (1.5 million, pre-activated via SIINFEKL (10μg/mL) for 72 h), and the tumor volume was recorded.

For in vivo CD8$^+$ T cell depletion, an anti-CD8 depletion antibody (5 mg/kg) was intraperitoneally injected on days 5, 8, and 11 after tumor inoculation, and the depletion effect was confirmed by flow cytometry.

## Statistics

Data were analyzed using the GraphPad Prism software, V.5. Comparisons between two groups were analyzed using a two-tailed unpaired Student's *t* test. Comparisons between multiple groups were analyzed using one-way analysis of variance (ANOVA) with Bonferroni's post-test or two-way ANOVA with Bonferroni's post-test for tumor growth study. Statistical significance was defined as a *P* value less than 0.05.

## Study approval

All mice were maintained under specific pathogen-free conditions and in accordance with the animal experimental guidelines of Sun Yat-sen University (Guangzhou, China). All animal procedures were approved by the Institutional Animal Care and Use Committee of Sun Yat-sen University (Guangzhou, China).

## Data availability

The raw data for RNA sequencing reported in this manuscript is available at Genome Sequence Archive (GSA) database of National Genomics Data Center (NGDC) with the accession number CRA007955/CRA018303 (www.ngdc.cncb.ac.cn/gsa). The data authenticity of this article has also been validated by uploading the key raw data onto the Research Data Deposit platform (www.researchdata.org.cn) and approved by the Sun Yat-sen University Cancer Center Data Access/Ethics Committee with the approval number RDDB2024200391.

The source data of this paper are collected in the following database record: biostudies:S-SCDT-10_1038-S44318-024-00217-y.

## Peer review information

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

## Acknowledgements

We thank Drs. Xinjian Li and Pengkai Sun at Institute of Biophysics, Chinese Academy of Sciences for key reagent support and helpful discussion. This work was supported by grants from the National Key R&D Program of China (2021YFC2400600/2021YFC2400601), the National Natural Science Foundation of China (82273189, 82073140, 81972692) and Fundamental Research Funds for the Central Universities (20ykzd22).Young Talents Program of Sun Yat-sen University Cancer Center (PT22270101).

## Author contributions

**Zining Wang**: Conceptualization; Funding acquisition; Investigation; Writing—original draft; Project administration; Writing—review and editing. **Lei Cui**: Data curation; Investigation. **Yanxun Lin**: Data curation. **Bitao Huo**: Formal analysis. **Hongxia Zhang**: Resources. **Chunyuan Xie**: Formal analysis. **Huanling Zhang**: Formal analysis. **Yongxiang Liu**: Methodology. **Huan Jin**: Investigation. **Hui Guo**: Data curation. **Mengyun Li**: Methodology. **Xiaojuan Wang**: Resources. **Penghui Zhou**: Resources. **Peng Huang**: Resources. **Jinyun Liu**: Resources. **Xiaojun Xia**: Conceptualization; Supervision; Funding acquisition; Project administration; Writing—review and editing.

Source data underlying figure panels in this paper may have individual authorship assigned. Where available, figure panel/source data authorship is listed in the following database record: biostudies:S-SCDT-10_1038-S44318-024-00217-y.

## Disclosure and competing interests statement

The authors declare no competing interests.

