## [Peer Review File · The EMBO Journal]

Tumor-intrinsic IRG1-itaconate axis promotes tumor immunogenicity

Zining Wang, Lei Cui, Yanxun Lin, Bitao Huo, Hongxia Zhang, Chunyuan Xie, Huanling Zhang, Yongxiang Liu, Huan Jin, Hui Guo, Mengyun Li, Xiaojuan Wang, penghui zhou, Peng Huang, Jinyun Liu, and Xiaojun Xia

Corresponding authors: Xiaojun Xia (xiaxj@sysucc.org.cn) , Zining Wang (wangzn@sysucc.org.cn)

Review Timeline:

Submission Date:	30th Jan 24
Editorial Decision:	28th Feb 24
Appeal:	8th May 24
Editorial Decision:	4th Jul 24
Revision Received:	16th Jul 24
Accepted:	7th Aug 24

Editor: Daniel Klimmeck

Transaction Report:

Dear Dr Xia,

Thank you for submitting your manuscript EMBOJ-2024-116840 for consideration by the EMBO Journal. Your manuscript has now been seen by three referees with expertise in tumor immunity and cancer metabolism, and we have received comments from all them which are shown below. In light of their comments, I am afraid we decided that we cannot offer publication in The EMBO Journal.

As you will see the experts state that the findings will as such be of interest to the field. At the same time, they also express substantial reservations with the analysis which in our view preclude further consideration by the EMBO Journal. These major concerns relate a diverse number of open points on the mechanism and signaling axis proposed (see referees #1 and #2), tumor cell specificity of the effects (ref#1) as well as important issues stated regarding claims made on endogenous itaconate and potentially distinct modes of action of itaconate derivatives (ref#2).

Given the negative opinions regarding the overall scientific advance and broader scope of the findings provided and considering the fact that the EMBO Journal can only afford to accept papers which receive enthusiastic support from a majority of referees, I am afraid we have concluded that we cannot offer to publish it here.

Thank you in any case for the opportunity to consider this manuscript. I am sorry we cannot be more positive on this occasion, but we hope nevertheless that you will find our referees' comments helpful.

Kind regards,

Daniel Klimmeck

Daniel Klimmeck, PhD
Senior Editor
The EMBO Journal

Referee #1:

This article demonstrates an unconventional role of tumor-intrinsic itaconate in triggering tumor immunogenicity in EG7 cells. The authors also discovered that thimerosal, a mercury-based preservative, could induce IRG1/itaconate expression in EG7 cells but not macrophages. Intratumoral thimerosal could sensitize responses toward anti-PD-1 therapy in the subcutaneous EG7 model. The concerns are as follows:

1. The rationale for using a 31-drug library to screen compounds that induce ITG expression was not explained clearly. Why did the authors specifically look into IRG expression in this small library? Also, the authors only partially checked itaconate, cis-aconitate, malate, and succinate in the TCA cycles in control vs. thimerosal-treated EG7 cells. Metabolomics screening of EG7 cells treated with or without thimerosal should be performed.
2. Although addressed in the discussion, the reason why thimerosal-induced IRG1/itaconate was "tumor-specific" was not very convincing. The RIPK2/IRF1 axis also exists in macrophages. TFEB could also trigger IRG1 expression and itaconate production in macrophages. Additionally, the reason why thimerosal only induced intracellular expression of itaconate was also not explained. Was that because of low levels of endogenous IRG1/itaconate in EG7 cells? If the concentration of thimerosal was increased, would extracellular itaconate be detected? Was itaconate detected in the extracellular fluid in subcutaneous EG7 tumors?
3. The authors demonstrated that thimerosal-induced ROS is the cause of upregulated RIPK3/IRF1 axis and subsequent increased expression of IRG1. However, ROS was known to induce production of various cytokines. Cytokine arrays should be performed to analyze potential cytokines induced by thimerosal.
4. The author used an in vitro T cell co-culture system to demonstrate the effect of thimerosal/itaconate on enhancing IFN γ production of antigen-specific CD8 T cells. However, a TME contains various TILs and all can be affected by thimerosal/itaconate treatment. A comprehensive analysis of TIL populations should be performed in vivo.
5. Translocation of TFEB to the nucleus does not mean that it binds to the promoter of B2m, Tap1, and Tap2. Thimerosal treatment enriches multiple transcription factors (Fig. S4A). TFEB is not one of the upregulated TFs. It is hard to believe that TFEB is the principal regulator of thimerosal/itaconate-induced tumor immunogenicity. In addition, how did thimerosal induce the nucleus translocation of TFEB?
6. The in vivo dosage of thimerosal is troublesome. In the animal model, the authors used intratumoral injection of thimerosal at a dose of 10 mg/kg, which was translated to 200 μ g per mouse. The FDA approved the use of thimerosal at a concentration of

0.003% to 0.01%. A vaccine containing 0.01% thimerosal as a preservative contains 50 µg of thimerosal per 0.5 ml dose. The authors have to explain why such a high dose was chosen in vivo. Plus, the growth inhibition effect of thimerosal could simply be due to the tumoricidal effect or inflammation induced by it.

Referee #2:

In this paper by Wang et al the authors propose that tumor-intrinsic itaconate production promotes an anti-tumor immune response. The authors propose that thimerosal, a vaccine preservative, induces IRG1 expression in tumor cells but not in macrophages, and that this enhances tumor immunogenicity by activating antigen presentation via TFEB. The authors propose that thimerosal promotes itaconate production through induction of ROS, RIPK3 and IRF1 and that inhibition of IRG1 impaired the anti-tumor effects of thimerosal. If the authors reworked this paper to focus on itaconate derivatives as a potential approach to increase antigen presentation in tumor cells this manuscript would be strengthened. Question marks about the significance of itaconate on tumor cells and the endogenous production of itaconate in tumor cells dampens my enthusiasm.

1. The authors need to distinguish between itaconate and itaconate derivatives such as 4OI and DMI. It is now well-appreciated that the mechanism of action of 4OI, DMI and itaconate differ substantially.
2. Itaconate acid is incorrect. This should read itaconate or itaconic acid.
3. How did the authors select 0.5 and 1 mM concentrations of itaconate? These are levels that are achieved intracellularly in macrophages. Extracellular itaconate concentrations have been reported to be in the low uM range. For example, see PMID: 30181801. Does itaconate have any effect on tumor cells at physiological concentrations that a tumor cell may be exposed to?
4. Furthermore, the lack of effect of itaconate absent digitonin suggests that exogenous itaconate is unlikely to alter tumor cell function.
5. The figure legend in Figure 2 is lacking. What do lanes 1-30 contain? What are the cell treatments?
6. As tumor cells express TLR4 why does TLR4 activation not affect IRG1 in tumor cells.
7. There appears to be a double band in IRG1 blots from tumor cells but not BMDMs. What is this band? The authors need to include a blot where tumor cells and BMDMs treated +/- LPS +/- thimerosal are on the same gel.
8. The same is true for itaconate production as in 7. The induction in tumor cells look negligible.
9. This is not a justified statement:
"We next treat tumor cells with thimerosal in combination with 4-OI. The results show that the combination treatment did not further increase tumor immunogenicity, suggesting that itaconate induction mediates thimerosal effect (Figure 2J, Figure S2H)." This could simply mean 4-OI has no effect.
10. The authors claim that itaconate induced tumor immunogenicity by up-regulating antigen presentation via TFEB but all experiments are performed with 4OI. Does itaconic acid affect TFEB or antigen presentation?
11. TLR4 is capable of inducing IRF1 which questions why LPS cannot boost IRG1 in tumor cells? This is true too for ROS and RIPK.
12. The effect of IRG1 ablation on tumor growth seems to vary across the 2 KD cells with one showing a phenotype and one not. This should be repeated for reproducibility.
13. Is there any difference in MHC expression, antigen presentation or T cell activity in mice injected with WT or IRG1KO tumor cells?
14. There are many typos and grammatical errors throughout.

Referee #3:

The paper by Wang et al describes the role of IRG1, key enzyme for itaconate metabolism, in cancer cells. They show that increased itaconate levels in cancer cells can boost the antigenicity of cancer cells leading to a better T cell activation both in vitro and in vivo. Moreover, they found that thimerosal, a vaccine preservative, can be used to pharmacologically induce IRG1 specifically in cancer cells.

Although growing literature describes the role of itaconate in the tumor microenvironment and in tumor progression, very few is known about the role of IRG1 in cancer cells. As such, the reviewer believes that the data are novel in the field of cancer immunology with interesting therapeutic potential.

The paper is well-written and experiments are generally well performed. Despite this, the Reviewer believes that the following comments need to be addressed before acceptance in EMBO Journal:

Major:

- 1) The in vivo data are very impressive, in particular considering the fact that the injection of thimerosal is performed only once and at an early time point. Although the Fig. 1E is very convincing showing the importance of Irg for the reduction in tumor volume, the Reviewer believes that the authors should show that the thimerosal treatment leads to an upregulation of Irg1 in the tumor setting (by for example by sorting CD45- cells or by immunohistochemistry). In this way, the authors could also show the levels of Irg1 in vivo in tumor cells that might differ from the levels in vitro. It would be also nice (but not essential) to check that in the immune compartment there is no induction of Irg upon treatment. Moreover, could the author speculate regarding how only one injection might be sufficient to observe such a strong effect in relatively long term?

- 2) In the discussion section, the authors said that "Our data show that tumor-derived itaconate production induced by thimerosal is strong enough to trigger tumor immunogenicity, but may not induce immune suppression in tumor microenvironment as it didn't secrete to extracellular space from tumor cells." and "As we couldn't detect extracellular itaconate in the culture media of tumor cells after thimerosal treatment, tumor-derived itaconate by thimerosal would most likely stay inside the cells for enhancing tumor immunogenicity rather than be released into tumor microenvironment and inducing immune suppression." Based on all the literature data regarding the role of secreted itaconate in the tumor microenvironment, the reviewer thinks that it is a very interesting and important data. The authors should show the data (itaconate levels intracellular vs extracellular) in the manuscript together with a positive control to be sure that the assay is well-performed and exclude any technical issue related to the no detection of itaconate in the medium. Moreover, itaconate levels at different time-points should be analyzed to be sure of the consistency of these observations.
- 3) Regarding FS3I and 3J. Could the authors speculate about the reason why the overexpression of TFEB does not lead to any increase in the control condition of B2m and LacZ activity? This condition (overexpression of TFEB) should mimic the treatment with 4-OI or thimerosal.
- 4) The authors should show a dose-dependent increase of Irg1 in cancer cells (in vitro) upon treatment with thimerosal to further validate the induction of Irg1 by thimerosal.
- 5) The authors should perform a cell-cycle and apoptosis analysis in vitro upon treatment with thimerosal and 4-OI (effect of Irg1 on cancer cell proliferation was previously reported PMC10077032) to be sure that the in vitro assays are not influenced by differences in number or states of the cancer cells.
- 6) What about the GO terms enriched in the RNA seq of thimerosal treated cells?
- 7) The fact that all the experiments with thimerosal are performed with EG7 should be briefly mentioned in the discussion: the authors should not generalize that this is a general mechanism (or treatment) of cancer cells. Further investigation in this line is needed.
- 8) Fig 4D,E, F and S3 G,I and J: not clear the statistic to which comparisons is referred to (the authors should show the statistic as in panel H). In addition, in shCTRL there is nothing significant in treatment vs DMSO?
- 9) Fig. 4H: is there any difference upon treatment in the shSCR?
- 10) How was performed the co-culture of cancer cells pre-treated and the T cells? How the authors exclude a direct effect of the treatment on the T cells? Were the cells washed?

Minor points:

- 11) The paper <https://doi.org/10.1038/s41467-023-43988-4> should be mentioned since it is in the field of itaconate and cancer.
- 12) In some panels the control condition is indicated as DMSO in others as NT. Is it a mistake of the labeling or one of the two conditions is used as control in different experiments?
- 13) Fig 2A. The reviewer believes that the # symbol is missing in the labeling of the WB.
- 14) Fig 5C. The quantification of the FACS is missing (something like Fig. 5 K should be shown).
- 15) Lysosome pathway is mainly involved in antigen presentation on MHC-II in immune cells rather than MHC-I that is more related to endosomal pathway (at least to the best of Reviewer knowledge).

** As a service to authors, EMBO Press provides authors with the possibility to transfer a manuscript that one journal cannot offer to publish to another EMBO publication or the open access journal Life Science Alliance launched in partnership between EMBO Press, Rockefeller University Press and Cold Spring Harbor Laboratory Press. The full manuscript and if applicable, reviewers' reports, are automatically sent to the receiving journal to allow for fast handling and a prompt decision on your manuscript. For more details of this service, and to transfer your manuscript please click on Link Not Available. **

Dear Dr. Klimmeck,

Thank you and the reviewers for the critical and helpful comments on our manuscript (EMBOJ-2024-116840). We have carefully read all the comments and performed additional experiments to address these questions and concerns. We believe that our new data has strongly substantiated our findings, and we would like to respectfully ask for an opportunity to have our revised manuscript re-assessed for potential publication in your esteemed journal. Below is the point-to-point response letter that we addressed each comment from the reviewers with new experiment results. Please kindly let us know if you would reconsider the revised manuscript, and inform us if you have any questions or need any further information.

Thank you very much and look forward to your favorable response!

Best Regards

Xiaojun Xia, Ph.D.

Investigator,

Dept. of Experiment Medicine

Sun Yat-sen University Cancer Center

Guangzhou, China 510060

Email: xiaxj@sysucc.org.cn

Below is the point-to-point response to referees' questions and concerns:

Referee #1:

This article demonstrates an unconventional role of tumor-intrinsic itaconate in triggering tumor immunogenicity in EG7 cells. The authors also discovered that thimerosal, a mercury-based preservative, could induce IRG1/itaconate expression in EG7 cells but not macrophages. Intratumoral thimerosal could sensitize responses toward anti-PD-1 therapy in the subcutaneous EG7 model. The concerns are as follows:

1. The rationale for using a 31-drug library to screen compounds that induce ITG expression was not explained clearly. Why did the authors specifically look into IRG expression in this small library? Also, the authors only partially checked itaconate, cis-aconitate, malate, and succinate in the TCA cycles in control vs. thimerosal-treated EG7 cells. Metabolomics screening of EG7 cells treated with or without thimerosal should be performed.

Response: Thanks for your comments and suggestions. We previously screened a 1080-drug library for immunogenic tumor cell death-inducing drugs (1). During the screening, we also found several drugs that inhibited cell proliferation and enhanced tumor immunogenicity with or without inducing cell death. In this study, we initially discovered the effect of itaconate and itaconate derivatives on triggering tumor immunogenicity, while IRG1 is the key enzyme for itaconate production, therefore, we focus on searching the IRG1-inducing drugs. The 31 drugs were pre-selected drugs that induce cell death or proliferation inhibition in our previous study(1), as we expected this IRG1 inducer could inhibit tumor growth not only depending on direct cell proliferation inhibition but also triggering tumor immunogenicity. As you suggested, we performed metabolomics screening of EG7 cells treated with or without thimerosal and found that the thimerosal treatment changed many metabolites in EG7 cells. Among these metabolites, itaconate was significantly induced by thimerosal treatment (**Response Figure 1**). We have incorporated the data and method into **Revised Figure S2C and Methods**.

Figure for reviewers removed

2. Although addressed in the discussion, the reason why thimerosal-induced IRG1/itaconate was "tumor-specific" was not very convincing. The RIPK2/IRF1 axis also exists in macrophages. TFEB could also trigger IRG1 expression and itaconate production in macrophages. Additionally, the reason why thimerosal only induced intracellular expression of itaconate was also not explained. Was that because of low levels of endogenous IRG1/itaconate in EG7 cells? If the concentration of thimerosal was increased, would extracellular itaconate be detected? Was itaconate detected in the extracellular fluid in subcutaneous EG7 tumors?

Response: Thanks for these important questions. We used an increasing dosage of thimerosal to treat BMDMs and found that the thimerosal treatment would induce BMDM death even at very low concentrations, which indicated that thimerosal prefers to induce cell death but not IRG1 expression in macrophages (**Response Figure 2A-B**). A recent report shows TFEB triggering IRG1 expression and itaconate production in BMDMs(2). Our result shows that the KD TFEB in EG7 cells didn't influence the IRG1 expression induced by thimerosal treatment (**Response Figure 2C**). This result indicated that TFEB is not the major transcript factor responsible for thimerosal-induced IRG1 expression in EG7 cells, suggesting different IRG1-inducing mechanisms in EG7 cells versus BMDMs. Our results also showed that thimerosal only increased the endogenous IRG1/Itaconate production, but not the extracellular itaconate. This may be due to the expression level of ABCG2, an itaconate exporter, in EG7 cells being too low to export the itaconate into the extracellular space versus in BMDMs (3), (**Response Figure2D**). Moreover, increasing the dosage of thimerosal treatment didn't further increase IRG1 expression but increased cell death in EG7 cells (**Response Figure2E-F**). As suggested, we detected the itaconate levels in extracellular fluid in subcutaneous EG7 tumors that were administrated with intratumoral thimerosal or PBS but found no significant difference (**Response Figure2G**). We have incorporated these data in the **Response Figure 2D** into **Revised Figure S10A**.

Figure for reviewers removed

3. The authors demonstrated that thimerosal-induced ROS is the cause of

upregulated RIPK3/IRF1 axis and subsequent increased expression of IRG1. However, ROS was known to induce production of various cytokines. Cytokine arrays should be performed to analyze potential cytokines induced by thimerosal.

Response: Thanks for your suggestion. Following your suggestion, we performed the cytokine array in EG7 cells which were treated with thimerosal or PBS. The result shows that among 11 cytokines, only IL4, IL18, and TNF α were upregulated by thimerosal treatment (**Response Figure3**). IL4 is generally considered a pro-cancer cytokine by inducing immunosuppressive tumor-promoting myeloid cells(4). IL18 can enhance NK cells to produce more IFN γ , while the IL18R α /R β is highly expressed in NK cells but not CD8 $^+$ T cells (5). TNF α is a classical inflammatory cytokine that may induce inflammation and trigger cell death but not enhance tumor immunogenicity. As our tumor-T cell co-culture system shows the thimerosal-treated tumor cells (but not cell culture supernatant) activate CD8 $^+$ T cells, we think these cytokines induced by thimerosal are unlikely responsible for the enhanced tumor immunogenicity.

Figure for reviewers removed

4. The author used an in vitro T cell co-culture system to demonstrate the effect of thimerosal/itaconate on enhancing IFN γ production of antigen-specific CD8 T cells. However, a TME contains various TILs and all can be affected by thimerosal/itaconate treatment. A comprehensive analysis of TIL populations should be performed in vivo.

Response: Thanks for your comments. Yes, we agree that in vivo administration of thimerosal would affect many types of cells in the TME. Following your suggestion, we have re-analyzed the TIL populations in PBS or thimerosal-treated EG7 tumors. In thimerosal-treated EG7 tumor tissues, CD8 $^+$ T cells were increased, CD11b $^+$ F4/80 $^+$ macrophages were decreased, and the B cells, NK1.1 and CD11b $^+$ CD11c $^+$ dendritic cells show no significant change (**Response Figure 4A-F**). Moreover, the activation of CD8 $^+$ T but not CD4 $^+$ T cells was also significantly increased (**Response Figure 4G-N**). We have incorporated the data in the **Response Figure 4** into **revised Figure 6** and **Figure S8**.

Figure for reviewers removed

5. Translocation of TFEB to the nucleus does not mean that it binds to the promoter of *B2m*, *Tap1*, and *Tap2*. Thimerosal treatment enriches multiple transcription factors (Fig. S4A). TFEB is not one of the upregulated TFs. It is hard to believe that TFEB is the principal regulator of thimerosal/itaconate-induced tumor immunogenicity. In addition, how did thimerosal induce the nucleus translocation of TFEB?

Response: Thanks for your comments and suggestions. To confirm if nuclear translocation of TFEB would bind to the promoter of target genes, we performed a ChIP-qPCR assay and found that thimerosal or 4-OI treatment increased TFEB binding on the promoter regions of *B2m*, *Tap1*, and *Tap2* (**Response Figure 5A**). Our previous results suggested that thimerosal triggered IRG1 expression and itaconate production, and then promoted nuclear translocation of TFEB to induce gene expression. To figure out why TFEB is not one of the upregulated TFs induced by thimerosal treatment in our RNA-seq data, we first checked the time course of IRG1 expression induced by thimerosal treatment in EG7 cells and found that IRG1 could only be detected after 12hrs of thimerosal treatment (**Response Figure 5B**). In contrast, the nuclear translocation of TFEB was also only detected after 18 hrs of thimerosal treatment (**Response Figure 5C**). As our previous RNA-seq samples were collected 12 hrs after thimerosal treatment, it may explain why we only detected the ROS response and transcription upregulation of IRF1 but not TFEB. Moreover, when IRG1 was knocked out in EG7 cells, the nuclear translocation of TFEB induced by thimerosal was completely inhibited (**Response Figure 5D**). A TFEB C270S mutation (which abolished its itaconate modification, as reported in reference (6)) attenuated its nuclear translocation induced by thimerosal (**Response Figure 5E**). These results suggested that thimerosal triggers IRG1/Itaconate production, which then modifies the TFEB for nuclear translocation and inducing gene expression. We have incorporated the data in the **Response Figure 5** into **revised Figure 3A, C**, and **Figure S3F-G**.

Figure for reviewers removed

6. The *in vivo* dosage of thimerosal is troublesome. In the animal model, the authors used intratumoral injection of thimerosal at a dose of 10 mg/kg, which was translated to 200 µg per mouse. The FDA approved the use of thimerosal at a concentration of 0.003% to 0.01%. A vaccine containing 0.01% thimerosal as a preservative contains 50 µg of thimerosal per 0.5 ml dose. The authors have to explain why such a high dose was chosen *in vivo*. Plus, the growth inhibition effect of thimerosal could simply be due to the tumoricidal effect or inflammation induced by it.

Response: Thank you for this important question. For the high dose we used for tumor treatment, we think that different applications of thimerosal require different concentrations. The FDA approved thimerosal for vaccine preservation for anti-bacterial purposes, and the concentration of 0.003% to 0.01% is enough to inhibit bacterial growth. In this study, we used thimerosal to trigger tumor immunogenicity in the tumor, and based on the dosage we used *in vitro* for cell treatment (10µg/mL), we chose the 10mg/kg *in vivo*. To avoid the potential side effects of thimerosal treatment, we deliver the drug via intra-tumoral injection. It is important to note that thimerosal only inhibited tumor growth in immune complement mice but not nude mice or CD8+T cell-depleted mice, indicating that the tumor inhibition by thimerosal was not dependent on the tumoricidal effect of high dose. Our result also showed that IRG1 KO in tumor cells also attenuated the tumor inhibition effect of thimerosal, further suggesting that tumor inhibition by thimerosal is neither dependent on tumoricidal effect nor inflammation.

Referee #2:

In this paper by Wang et al the authors propose that tumor-intrinsic itaconate production promotes an anti-tumor immune response. The authors propose that thimerosal, a vaccine preservative, induces IRG1 expression in tumor cells but not in macrophages, and that this enhances tumor immunogenicity by activating antigen presentation via TFEB. The authors propose that thimerosal promotes itaconate production through induction of ROS, RIPK3 and IRF1 and that inhibition of IRG1 impaired the anti-tumor effects of thimerosal. If the authors reworked this paper to focus on itaconate derivatives as a potential approach to increase antigen presentation in tumor cells this manuscript would be strengthened. Question marks about the significance of itaconate on tumor cells and the endogenous production of itaconate in tumor cells dampens my enthusiasm.

1. The authors need to distinguish between itaconate and itaconate derivatives such as 4OI and DMI. It is now well-appreciated that the mechanism of action of 4OI, DMI and itaconate differ substantially.

Response: Thanks for the important note. Yes, we agree and we are aware that the itaconate and itaconate derivatives play different roles in some conditions, such as itaconate derivatives but not itaconate induce NRF2(7). Our new results show that the supplement of an NRF2 inhibitor (ML385) did not attenuate thimerosal- or 4-OI-triggered tumor immunogenicity (**Response Figure 6A**), indicating an NRF2-independent effect. Moreover, our co-culture model shows that itaconate derivatives and itaconate plus digitonin (digitonin could help itaconate entry into cells(6)) trigger tumor immunogenicity (**Response Figure 6B**). Itaconate plus digitonin could also mimic the 4-OI effect on inducing nuclear translocation of TFEB (**Response Figure 6C**). Our results also show that itaconate plus digitonin can enhance MHC-I and MHC-I SIINFEKL expression in EG7 cells, which is consistent with its derivatives (**Response Figure 6D**). Thus, we believe that itaconate and its derivatives play the same role in inducing tumor immunogenicity in our study. We have incorporated the data in the **Response Figure 6C** into **revised Figure S3E**.

Figure for reviewers removed

2. Itaconate acid is incorrect. This should read itaconate or itaconic acid.

Response: Thanks for pointing out this mistake. We have corrected this mistake in our revised manuscript.

3. How did the authors select 0.5 and 1 mM concentrations of itaconate?

These are levels that are achieved intracellularly in macrophages.

Extracellular itaconate concentrations have been reported to be in the low μM range. For example, see PMID: 30181801. Does itaconate have any effect on tumor cells at physiological concentrations that a tumor cell may be exposed to?

Response: Thanks for your comments and great questions. According to recent studies, the intracellular itaconate would be induced when treated with LPS for 4-6 hours(8, 9). Indeed, the extracellular itaconate concentration is low after 6hrs with LPS treatment, while the extracellular itaconate continuously accumulated and eventually comprised more than 90% of the newly synthesized itaconate, which would reach around $50\mu\text{M}$ after 24hrs(8). And recent studies showed that the work concentration of itaconate is around 1mM (10-12). So, we select 0.5 and 1 mM concentrations of itaconate in our study. Unexpectedly, we used an increased concentration of itaconate to treat EG7 cells, the results show itaconate can't trigger tumor immunogenicity (**Response Figure 7A**), which may be because the itaconate can't enter EG7 cells directly. While, we used itaconate combined with digitonin to treat EG7 tumor cells, as a recent report showed that digitonin could help itaconate entry into cells. We observed the itaconate (upon $150\mu\text{M}$) plus digitonin enhance tumor immunogenicity (**Response Figure 7B**)

Figure for reviewers removed

4. Furthermore, the lack of effect of itaconate absent digitonin suggests that exogenous itaconate is unlikely to alter tumor cell function.

Response: Thanks for your comments. Yes, our results show that treating EG7 cells with itaconate directly can't trigger tumor immunogenicity, while digitonin can help itaconate entry into EG7 cells to trigger tumor immunogenicity, which mimics the itaconate derivatives' function (**Response Figure 7A, 7B**). In this study, we used thimerosal to induce itaconate production inside the tumor cells for enhancing tumor immunogenicity, and cellular uptake of exogenous itaconate was not involved in this situation.

5. The figure legend in Figure 2 is lacking. What do lanes 1-30 contain? What are the cell treatments?

Response: Thanks for pointing out the mistake. We have included the figure legend in our revised Figure 2 and provided the details in our revised manuscript. The lanes of 1-30 present 30 different drugs (Supplementary Table 1) we used for treating the EG7 cells.

Table 1 The list of drugs for screening

Number	Drugs	Number	Drugs
1	Dactinomycin	16	Emetine Dihydrochloride
2	Methotrexate(+/-)	17	Quinidine Gluconate
3	Minoxidil	18	Sulfacetamide
4	Vinblastine Sulfate	19	Spiramycin
5	Ethyl Vanillin	20	Alrestatin
6	Docetaxel	21	Cyclosporine
7	Dasatinib	22	Monensin Sodium
8	Podofilox	23	Benzoic Acid
9	Acrisorcin	24	Benzyl Benzoate

10	Adenosine	25	Floxuridine
11	Chlorhexidine Dihydrochloride	26	Niclosamide
12	Cortisone Acetate	27	Naltrexone Hydrochloride
13	Dicloxacillin Sodium	28	Adapalene
14	Digitoxin	29	Metoclopramide Hydrochloride
15	Deflazacort	30	Mercaptopurine
		#	Thimerosal

6. As tumor cells express TLR4 why does TLR4 activation not affect IRG1 in tumor cells. *Figure for reviewers removed*

Response: Thanks for your question. We checked the expression level of TLR4 in EG7 and BMDMs by FACS analysis and found that the expression level of TLR4 is at a very low level in EG7 compared to that of macrophages (**Response Figure 8A**). When these cells were treated with LPS, phosphorylated-p65 was only increased in BMDMs but not EG7 cells (**Response Figure 8B**). These results indicated the TLR4 signaling is defective in EG7 cells, which may explain why LPS didn't induce IRG1 expression in EG7 cells.

Figure for reviewers removed

7. There appears to be a double band in IRG1 blots from tumor cells but not BMDMs. What is this band? The authors need to include a blot where tumor cells and BMDMs treated +/- LPS +/- thimerosal are on the same gel.

Response: Thanks for your comments and suggestions. We checked the IRG1 blots in EG7 WT and IRG1 KO cells (**Response Figure 9A**). We also included the blot of EG7 cells and BMDMs on the same gel (**Response Figure 9B**), and we concluded the band (around 50-Kd, red arrow) was a non-specific signaling.

Figure for reviewers removed

8. The same is true for itaconate production as in 7. The induction in tumor cells look negligible.

Response: Thanks for your comments and suggestions. We check the itaconate production of EG7 cells and BMDMs which were treated with LPS or thimerosal. Consistently to our previous observation, only thimerosal but not LPS induced itaconate production in EG7 cells. On the contrary, only LPS but not thimerosal induced itaconate production in BMDMs (**Response Figure 10**). Indeed, the LPS-induced itaconate level in macrophages is much higher than that of tumor cells induced by thimerosal. This also indicates that different itaconate levels induced by different upstream signaling lead to distinct biological functions in various cell types, such as the antibacterial function in BMDMs. Our data also show that tumor-derived itaconate production induced by thimerosal is strong enough to trigger tumor immunogenicity, as KO IRG1 in tumor cells abolishes thimerosal-induced itaconate production and tumor immunogenicity.

Figure for reviewers removed

9. This is not a justified statement:

"We next treat tumor cells with thimerosal in combination with 4-OI. The results show that the combination treatment did not further increase tumor immunogenicity, suggesting that itaconate induction mediates thimerosal effect (Figure 2J, Figure S2H)." This could simply mean 4-OI has no effect.

Response: Thanks for your comments. We repeat the experiments and add the 4-OI treatment alone group. Consistent with our previous observation, thimerosal and 4-OI can trigger tumor immunogenicity. Compared to the thimerosal treatment alone group, the combination treatment did not further increase tumor immunogenicity (**Response Figure 11**). We have incorporated the data in the Response Figure 11 into revised Figure 2J and Figure S2K.

Figure for reviewers removed

10. The authors claim that itaconate induced tumor immunogenicity by up-regulating antigen presentation via TFEB but all experiments are performed with 4OI. Does itaconic acid affect TFEB or antigen presentation?

Response: Thanks for your comments and suggestions. Our new results show itaconate plus digitonin can induce TFEB nuclear translocation (**Response Figure 11A**), accompanied by enhanced MHC-I and MHC-I SIINFEKL expression in EG7 cells (**Response Figure 11B**).

Figure for reviewers removed

11. TLR4 is capable of inducing IRF1 which questions why LPS cannot boost IRG1 in tumor cells? This is true too for ROS and RIPK.

Response: Thanks for your comments and suggestions. Our new results show the TLR4 signaling is defective in EG7 cells, which makes it unresponsive to LPS stimulation (**Response Figure 8A-B**). We also check the ROS in BMDMs and EG7 cells. The basal level of ROS in EG7 cells is much higher than that of BMDMs (**Response Figure 12A**). We also observed that BMDMs are more vulnerable to thimerosal-induced cell death (**Response Figure 12B**). Together, we reason that a high basal level of ROS in tumor cells makes EG7 more sensitive to ROS inducers such as thimerosal, thereby activating downstream signaling including RIPK3/IRF1 but not cell death.

Figure for reviewers removed

12. The effect of IRG1 ablation on tumor growth seems to vary across the 2 KD cells with one showing a phenotype and one not. This should be repeated for reproducibility.

Response: Thanks for your comments and suggestions. We have repeated this in vivo experiment. Both IRG1 KO lines showed attenuated tumor inhibition effect of thimerosal treatment (**Response Figure 13A-B**). We have incorporated the data in **Response Figure 13** into **revised Figure 6K-L**.

Figure for reviewers removed

13. Is there any difference in MHC expression, antigen presentation or T cell activity in mice injected with WT or IRG1KO tumor cells?

Response: Thanks for your comments and suggestions. We checked the intratumoral MHC-I and MHC-I SIINFEKL expression levels by FACS and found the levels in WT but not IRG1 KO tumors were increased after intra-tumoral thimerosal treatment (**Response Figure 14A-B**). And the expression of B2m and Tap1 shows a similar trend (**Response Figure 14C**). Moreover, thimerosal treatment increased T cell infiltration and activation in WT but not IRG1 KO tumors (**Response Figure 14D-G**). We have incorporated the data in the Response Figure 14 into revised Figure 6M-P and Figure S8N-P.

Figure for reviewers removed

14. There are many typos and grammatical errors throughout.

Response: Thanks for your comments. We apologize for the typos and grammatical errors. We have corrected the errors in our revised manuscript, and we also used an English language editing service to improve the writing style of the manuscript. We hope the writing will now satisfy the publication standard.

Referee #3:

The paper by Wang et al describes the role of IRG1, key enzyme for itaconate metabolism, in cancer cells. They show that increased itaconate levels in cancer cells can boost the antigenicity of cancer cells leading to a better T cell activation both in vitro and in vivo. Moreover, they found that thimerosal, a vaccine preservative, can be used to pharmacologically induce IRG1 specifically in cancer cells.

Although growing literature describes the role of itaconate in the tumor microenvironment and in tumor progression, very few is known about the role of IRG1 in cancer cells. As such, the reviewer believes that the data are novel in the field of cancer immunology with interesting therapeutic potential. The paper is well-written and experiment are generally well performed. Despite this, the Reviewer believe that the following comments need to be addressed before acceptance in EMBO Journal:

Major:

1) The *in vivo* data are very impressive, in particular considering the fact that the injection of thimerosal is performed only once and at an early time point. Although the Fig. 1E is very convincing showing the importance of Irg for the reduction in tumor volume, the Reviewer believes that the authors should show that the thimerosal treatment leads to an upregulation of Irg1 in the tumor setting (by for example by sorting CD45- cells or by immunohistochemistry). In this way, the authors could also show the levels of Irg1 *in vivo* in tumor cells that might differ from the levels *in vitro*. It would be also nice (but not essential) to check that in the immune compartment there is no induction of Irg upon treatment. Moreover, could the author speculate regarding how only one injection might be sufficient to observe such a strong effect in relatively long term?

Response: Thanks for your encouraging comments and suggestions. We sorted out the immune cells and non-immune cells from the PBS- or thimerosal-treated tumor tissues and examined IRG1 expression by WB. The result shows that thimerosal treatment leads to significant upregulation of IRG1 expression in non-immune cells but not immune cells (**Response Figure 15**). Our results show that thimerosal could trigger tumor immunogenicity *in vitro and in vivo*. Moreover, due to the upregulated immunogenicity induced by thimerosal treatment, more CD8+ T cells were infiltrated into tumor tissue and activated, which in turn inhibits tumor growth, activates tumor microenvironment, and boosts the host's anti-tumor immune response circle. We think this may be the reason why only one injection of thimerosal could be sufficient to inhibit tumor growth in a relatively long term. We have incorporated the data in **Response Figure 15** into revised **Figure S8B**.

Figure for reviewers removed

2) In the discussion section, the authors said that "Our data show that tumor-derived itaconate production induced by thimerosal is strong enough to trigger tumor immunogenicity, but may not induce immune suppression in tumor microenvironment as it didn't secret to extracellular space from tumor cells." and "As we couldn't detect extracellular itaconate in the culture media of tumor cells after thimerosal treatment, tumor-derived itaconate by thimerosal would most likely stay inside the cells for enhancing tumor immunogenicity rather than be released into tumor microenvironment and inducing immune suppression."

Based on all the literature data regarding the role of secreted itaconate in the tumor microenvironment, the reviewer thinks that it is a very interesting and important data. The authors should show the data (itaconate levels intracellular vs extracellular) in the manuscript together with a positive control to be sure that the assay is well-performed and exclude any technical issue related to the no detection of itaconate in the medium. Moreover, itaconate levels at different time-points should be analyzed to be sure of the consistency of these observation.

Response: Thanks for your encouraging comments and important suggestions. We have added the data of itaconate levels intracellular vs extracellular in our revised manuscript (Figure S2), and we used the medium supplemented with itaconate as the positive control (PC) (**Response Figure 16A**). We also checked the IRG1 expression and itaconate production in EG7 cells at different time points after thimerosal treatment. The results show IRG1 and itaconate are continuously accumulated in EG7 cells (**Response Figure 16B-C**).

Figure for reviewers removed

3) Regarding FS3I and 3J. Could the authors speculate about the reason why the overexpression of TFEB does not lead to any increase in the control

condition of B2m and LacZ activity? This condition (overexpression of TFEB) should mimic the treatment with 4-OI or thimerosal.

Response: Thanks for your question. Our result shows overexpression of TFEB didn't increase the B2m and LacZ activity as the treatment of 4-OI or thimerosal. We also noted that overexpressed TFEB was mainly in the cytoplasmic fraction, and thimerosal treatment enhanced its nuclear translocation (**Response Figure 5E**). Thus, overexpressed TFEB still needs further stimulation for nuclear translocation and gene induction. We speculate that without stimulation, most of the overexpressed TFEB would be likely to be phosphorylated by mTOR and bind with 14-3-3 to keep their cytosolic retention (6, 13). Once the cells were treated with 4-OI or thimerosal, the TFEB would be modified by 4-OI or itaconate and then translocated into nuclear and led gene expression. Future studies would be needed to delineate the detailed mechanism.

4) The authors should show a dose-dependent increase of Irg1 in cancer cells (in vitro) upon treatment with thimerosal to further validate the induction of Irg1 by thimerosal.

Response: Thanks for the great suggestion. We used an increased dosage of thimerosal to treat EG7 cells and found that thimerosal indeed induced IRG1 expression in a dose-dependent fashion (**Response Figure 17**).

Figure for reviewers removed

5) The authors should perform a cell-cycle and apoptosis analysis in vitro upon treatment with thimerosal and 4-OI (effect of Irg1 on cancer cell proliferation was previously reported PMC10077032) to be sure that the in vitro assays are not influenced by differences in number or states of the cancer cells.

Response: Thanks for your great suggestions. Following your suggestion, we used thimerosal or 4-OI treated with EG7 cells for 18 hrs and then checked the cell cycle (via PI staining) and apoptosis (via apoptosis assay kit) of EG7 cells by FACS. The results showed that thimerosal but not 4-OI increased G0/G1 population (**Response Figure 18A**), and thimerosal and 4-OI both induced very few cell apoptosis (**Response Figure 18B-C**). Moreover, we also found that only the live cells after thimerosal treatment could activate co-cultures T cells (**Response Figure 18D**). Taken together, we conclude that

thimerosal and 4-OI trigger tumor immunogenicity independent of the influence of cell cycle and cell apoptosis.

Figure for reviewers removed

6) What about the GO terms enriched in the RNA seq of thimerosal treated cells?

Response: Thanks for the question. We re-analyzed our RNA-seq data and found the top 10 of enriched GO terms in thimerosal-treated cells are associated with cell metabolism, stress, and ROS (**Response Figure 19**). This is also consistent with our metabolomics screening data.

Figure for reviewers removed

7) The fact that all the experiment with thimerosal are performed with EG7 should be briefly mentioned in the discussion: the authors should not generalize that this is a general mechanism (or treatment) of cancer cells. Further investigation in this line are needed.

Response: Thanks for the important suggestions. Our results show that thimerosal triggers tumor immunogenicity via increasing the ROS and activating the RIPK3-IRF1 axis to trigger IRG1 expression and itaconate production in EG7 cells. Indeed, due to the mutation or deficiency of this signaling axis in cancer cells, thimerosal can't induce IRG1 expression and itaconate production in all types of cancer cells. Thus, only those cancer cells with a complete signaling pathway axis may benefit from the thimerosal treatment. We next plan to check the other cell lines which have complete signaling pathway axis, and use the cell lines to verify the IRG1-itaconate axis in tumor cells promoting tumor immunogenicity *in vitro* and *in vivo*. We have included the possibility into the revised discussion part.

8) Fig 4D,E, F and S3 G,I and J: not clear the statistic to which comparisons is referred to (the authors should show the statistic as in panel H). In addition, in shCTRL there is nothing significant in treatment vs DMSO?

Response: Thanks for your questions. We have added the detailed information in our revised Figures 3 and 4. Our previous statistic comparisons focused on the difference between the shCTRL and shTFEB. We re-analyzed the data, and the new results showed the 4-OI and thimerosal treatment increased the tumor immunogenicity significantly in shCTRL cells.

9) Fig. 4H: is there any difference upon treatment in the shSCR?

Response: Thanks for your comments and suggestions. We performed statistical analysis on the data, and the 4-OI or thimerosal treatment increased the IFN γ and GzmB expression in the co-cultured OT-I cells significantly. We have added the results in our revised Figure 3H.

10) How was performed the co-culture of cancer cells pre-treated and the T cells? How the authors exclude a direct effect of the treatment on the T cells? Were the cells washed?

Response: Thanks for your comments. The treatment of 4-OI, DI, or thimerosal was removed by refreshing media before co-culture with T cells. We also tested the condition when these treatments were kept in the media throughout the experiment. The results show that presented DI and thimerosal but not 4-OI could still induce co-cultured T cell activation, but weaker than the removed group (**Response Figure 20**). These data

suggested that these drugs mainly act on tumor cells and enhance tumor immunogenicity to induce co-cultured T cell activation.

Figure for reviewers removed

Minor points:

11) The paper <https://doi.org/10.1038/s41467-023-43988-4> should be mentioned since it is in the field of itaconate and cancer.

Response: Sorry for the omission of this most recent important work. We have now cited and briefly discussed the findings in this reference in our revised manuscript (Reference #22).

12) In some panels the control condition is indicated as DMSO in others as NT. Is it a mistake of the labeling or one of the two condition is used as control in different experiment?

Response: Thanks for pointing out the mistakes in labeling. All the controls in our experiment are supplied with either DMSO or PBS. We have corrected this mistake and specified the control reagent in our revised figures.

13) Fig 2A. The reviewer believes that the # symbol is missing in the labeling of the WB.

Response: Sorry for the error. We have added the # symbol in our revised Figure 2A.

14) Fig 5C. The quantification of the FACS is missing (something like Fig. 5 K should be shown).

Response: Thanks for the comment. We have added the quantification data in our revised Figure 5C.

15) Lysosome pathway is mainly involved in antigen presentation on MHC-II in immune cells rather than MHC-I that is more related to endosomal pathway (at least to the best of Reviewer knowledge).

Response: Thanks for your very insightful comment. Indeed, the lysosome pathway is generally involved in antigen presentation on MHC-II in immune cells, and a previous study reported TFEB in regulating the MHC-II pathway in dendritic cells(14). Meanwhile, there is also a recent report showing that overexpression of TFEB in Merkel cell carcinoma cells upregulated the HLA-ABC (human MHC-I molecule) expression and consequent tumor antigenicity (ranked top 8 in 71 candidates hit in their screening for factors that upregulates HLA-I expression in Fig. 4) (15). In our study, we demonstrated that itaconate-induced TFEB also upregulated antigen presentation to enhance the tumor immunogenicity in EG7 cells. We also observed the lysosome pathway was enriched in itaconate-treated EG7 cells. TFEB is a multifunctional factor involved in lysosome biogenesis as well as autophagy, inflammation, antiviral responses, etc (16). The TFEB likely plays different roles in regulating antigen presentation in different cell types with different partners (e.g. LPS induced TFEB expression but not its partner TFE3 expression in macrophages (17)), or TFEB transcription function activated by itaconate may induce both MHC-I antigen presentation molecule transcription and lysosome pathway with different kinetics in EG7 cells, thereby guarantee its distinct functions in regulating different pathways. Future studies would be warranted to delineate the tumor-specific antigen processing and immunogenicity regulation by itaconate.

References

1. Z. Wang *et al.*, cGAS/STING axis mediates a topoisomerase II inhibitor-induced tumor immunogenicity. *J Clin Invest* **129**, 4850-4862 (2019).
2. E. M. Schuster *et al.*, TFEB induces mitochondrial itaconate synthesis to suppress bacterial growth in macrophages. *Nat Metab* **4**, 856-866 (2022).
3. C. Chen *et al.*, ABCG2 is an itaconate exporter that limits antibacterial innate immunity by alleviating TFEB-dependent lysosomal biogenesis. *Cell Metab* **36**, 498-510 e411 (2024).
4. N. M. LaMarche *et al.*, An IL-4 signalling axis in bone marrow drives pro-tumorigenic myelopoiesis. *Nature* **625**, 166-174 (2024).
5. T. Zhou *et al.*, IL-18BP is a secreted immune checkpoint and barrier to IL-18 immunotherapy. *Nature* **583**, 609-614 (2020).
6. Z. Zhang *et al.*, Itaconate is a lysosomal inducer that promotes antibacterial innate immunity. *Mol Cell* **82**, 2844-2857 e2810 (2022).
7. A. Swain *et al.*, Comparative evaluation of itaconate and its derivatives reveals divergent inflammasome and type I interferon regulation in macrophages. *Nat Metab* **2**, 594-602 (2020).
8. Y. R. Zeng *et al.*, The immunometabolite itaconate stimulates OXGR1 to promote mucociliary clearance during the pulmonary innate immune response. *J Clin Invest* **133**,

- (2023).
9. J. Meiser *et al.*, Itaconic acid indicates cellular but not systemic immune system activation. *Oncotarget* **9**, 32098-32107 (2018).
 10. H. Zhao *et al.*, Myeloid-derived itaconate suppresses cytotoxic CD8(+) T cells and promotes tumour growth. *Nat Metab* **4**, 1660-1673 (2022).
 11. M. Bambouskova *et al.*, Itaconate confers tolerance to late NLRP3 inflammasome activation. *Cell Rep* **34**, 108756 (2021).
 12. M. C. Runtsch *et al.*, Itaconate and itaconate derivatives target JAK1 to suppress alternative activation of macrophages. *Cell Metab* **34**, 487-501 e488 (2022).
 13. R. Puertollano, S. M. Ferguson, J. Brugarolas, A. Ballabio, The complex relationship between TFEB transcription factor phosphorylation and subcellular localization. *EMBO J* **37**, (2018).
 14. M. Samie, P. Cresswell, The transcription factor TFEB acts as a molecular switch that regulates exogenous antigen-presentation pathways. *Nat Immunol* **16**, 729-736 (2015).
 15. P. C. Lee *et al.*, Reversal of viral and epigenetic HLA class I repression in Merkel cell carcinoma. *J Clin Invest* **132**, (2022).
 16. O. A. Brady, J. A. Martina, R. Puertollano, Emerging roles for TFEB in the immune response and inflammation. *Autophagy* **14**, 181-189 (2018).
 17. N. Pastore *et al.*, TFEB and TFE3 cooperate in the regulation of the innate immune response in activated macrophages. *Autophagy* **12**, 1240-1258 (2016).

Dear Dr Xia,

Thank you for submitting your revised manuscript (EMBOJ-2024-116840R-Q) to The EMBO Journal. As mentioned, your amended study and the appeal response were sent back to the three referees for their re-evaluation, and we have received comments from all of them, which I enclose below. As you will see, the experts stated that the work has been substantially improved by the revisions and they are now in favour of publication, pending a final revision.

Thus, we are able to invite you to revise your manuscript according to the remaining issues raised. Upon amendment we will then be prepared to work towards acceptance and publication in The EMBO Journal.

Please consider the remaining points by referees #2 and #3 carefully, amending the study with additional experiments and adjusting the text where appropriate.

Also, we now need you to take care of a number of minor issues related to formatting and data presentation as detailed below, which should be addressed at re-submission.

Please contact me at any time if you have additional questions related to below points.

As you might have seen on our web page, every paper at the EMBO Journal now includes a 'Synopsis', displayed on the html and freely accessible to all readers. The synopsis includes a 'model' figure as well as 2-5 one-short-sentence bullet points that summarize the article. I would appreciate if you could provide this figure and the bullet points.

Thank you for giving us the chance to consider your manuscript for The EMBO Journal. I look forward to your final revision.

Again, please contact me at any time if you need any help or have further questions.

Kind regards,

Daniel Klimmeck

>> Please add maximally five keywords to the study.

>> Author Contributions: Please remove the author contributions information from the manuscript text. Note that CRediT has replaced the traditional author contributions section as of now because it offers a systematic machine-readable author contributions format that allows for more effective research assessment. and use the free text boxes beneath each contributing author's name to add specific details on the author's contribution.

More information is available in our guide to authors.
<https://www.embopress.org/page/journal/14602075/authorguide>

>> Adjust the title of the 'Conflict of Interest' section to 'Disclosure and Competing Interests Statement'.

>> Provide the main manuscript text as .doc file.

>> Figure callouts: Figure 4D needs to be called out.

>> Please provide source data for the study as to the separate request e-mail by my colleague Hannah Sonntag.

>> Appendix formatting: The file should be renamed "Appendix" and needs a table of contents (with page numbers) on its first page; correct nomenclature is "Appendix Figure S1" etc. The table will also need to be renamed "Appendix Table S1".

>> Figures in separate files: Figures should be removed from the manuscript and uploaded as individual, high resolution figure files. The legends should be compiled at the end of the manuscript text, after the References.

>> Data availability section: please enter a Data availability section into the manuscript. Detail the metabolomics analysis and make the data set publicly available.

>> Reference list format: please adjust to EMBO Journal format and correct to 10 author names listed before et al. .

>> Funding: information on funding is incomplete in our online system. All funders, including project numbers, need to be entered into our system in addition to being mentioned in the Acknowledgements.

>> Author checklist: provide a completed author checklist for your study.

>> Add a completed Reagents and Tools table to the Methods.

>> Consider additional changes and comments from our production team as indicated below:

- DAS:

Please note that the data availability statement is not provided in the manuscript.

- Figure legends:

1. Please note that the exact p values are not provided in the legends of figures 1a-g; 2d-k, m-n; 3b-i; 4c, f-h, l-n; 5c-d, g-l; 6b, d-r.
2. Please note that in figures 2d-k, m-n; 4g; 5c-d, g-l; there is a mismatch between the annotated p values in the figure legend and the annotated p values in the figure file that should be corrected.
3. Please note that information related to n is missing in the legends of figures 1a-g; 2d-k, m-n; 3b-i; 4c, f-h, l-n; 5c-d, g-l, 6k-l.
4. Please note that the error bars are not defined in the legends of figures 1a-g; 2d-k, m-n; 3b-i; 4c, f-h, l-n; 5c-d, g-l.

Referee #1:

The authors addressed my concerns.

Referee #2:

The authors have improved their manuscript but there are still some issues.

1. The figure legends still lack sufficient details to interpret them properly e.g. concentrations of drugs.
2. Some of the figures that are in the revision file should be included as supplementary data. This includes Response Figure 7, Response Figure 9-11.
3. The revised in vivo data improves the manuscript.

Referee #3:

The paper by Wang et al describes how the increase of IRG1, key enzyme for itaconate metabolism, can boost the antigenicity of cancer cells leading to a better anti-tumoral T cell response both in vitro and in vivo. Moreover, they described how thimerosal, a vaccine preservative, can be used to therapeutically induce IRG1 specifically in cancer cells and how this treatment can synergize with immunotherapy.

Although growing literature describes the role of itaconate in the tumor microenvironment, the role of IRG1 in cancer cells is still almost totally elusive. For this reason, combined with the strong efficacy of thimerosal in reducing tumor growth, the reviewer believes that the data are interesting and novel in the field of cancer immunology and cancer therapy.

Moreover, compared with the previous version of the manuscript assessed by the present Reviewer there is a better elucidation of the role of itaconate vs its derivatives. In parallel, additional experiments provided further clues to understand why the treatment with thimerosal induces Irg1 specifically in cancer cells and not in macrophages. Finally, a better description of the molecular pathway induced by thimerosal treatment and additional in vivo analysis have been performed.

The paper is well-written and the experiments are generally well performed and the message of the manuscript is strengthened in the current version. Despite this, the Reviewer believes that few minor comments need to be addressed before acceptance in EMBO Journal:

1) Response: Thanks for your encouraging comments and suggestions. We sorted out the immune cells and non-immune cells from the PBS- or thimerosal-treated tumor tissues and examined IRG1 expression by WB. The result shows that thimerosal treatment leads to significant upregulation of IRG1 expression in non-immune cells but not immune cells (Response Figure 15). Our results show that thimerosal could trigger tumor immunogenicity in vitro and in vivo. Moreover, due to the upregulated immunogenicity induced by thimerosal treatment, more CD8+ T cells were infiltrated into tumor tissue and activated, which in turn inhibits tumor growth, activates tumor microenvironment, and boosts the host's anti-tumor immune response circle. We think this may be the reason why only one injection of thimerosal could be sufficient to inhibit tumor growth in a relatively long term. We have incorporated the data in Response Figure 15 into revised Figure S8B.

 The data are very convincing (although the quality of the WB for the immune cells should be improved). Despite this, some information is missing, in particular which one was the gating strategy to sort the cells? Did the authors use CD45+ vs CD45-?

2) Thanks for your encouraging comments and important suggestions. We have added the data of itaconate levels intracellular vs extracellular in our revised manuscript (Figure S2), and we used the medium supplemented with itaconate as the positive control (PC) (Response Figure 16A). We also checked the IRG1 expression and itaconate production in EG7 cells at different time points after thimerosal treatment. The results show IRG1 and itaconate are continuously accumulated in EG7 cells (Response Figure 16B-C).

 The data are convincing. Despite this, I would perform a later time-point such as 48 or 72 hours to check both intracellular and, more importantly, extracellular levels to be sure that itaconate is not exported when it started to reach relatively "high intracellular levels"

Moreover, based on the importance of extracellular itaconate, the Reviewer would suggest to put the panel A in the main figure.

3) Response: Thanks for your question. Our result shows overexpression of TFEB didn't increase the B2m and LacZ activity as the treatment of 4-OI or thimerosal. We also noted that overexpressed TFEB was mainly in the cytoplasmic fraction, and thimerosal treatment enhanced its nuclear translocation (Response Figure 5E). Thus, overexpressed TFEB still needs further stimulation for nuclear translocation and gene induction. We speculate that without stimulation, most of the overexpressed TFEB would be likely to be phosphorylated by mTOR and bind with 14-3-3 to keep their cytosolic retention (6, 13). Once the cells were treated with 4-OI or thimerosal, the TFEB would be modified by 4-OI or itaconate and then translocated into nuclear and led gene expression. Future studies would be needed to delineate the detailed mechanism.

 Thank you for the answer, the proposed speculation sounds reasonable.

4) Thanks for the great suggestion. We used an increased dosage of thimerosal

to treat EG7 cells and found that thimerosal indeed induced IRG1 expression in a dose-dependent fashion (Response Figure 17).

 Thank you, very convincing data.

5) Thanks for your great suggestions. Following your suggestion, we used thimerosal or 4-OI treated with EG7 cells for 18 hrs and then checked the cell cycle (via PI staining) and apoptosis (via apoptosis assay kit) of EG7 cells by FACS. The results showed that thimerosal but not 4-OI increased G0/G1 population (Response Figure 18A), and thimerosal and 4-OI both induced very few cell apoptosis (Response Figure 18B-C). Moreover, we also found that only the live cells after thimerosal treatment could activate co-cultured T cells (Response Figure 18D). Taken together, we conclude that thimerosal and 4-OI trigger tumor immunogenicity independent of the influence of cell cycle and cell apoptosis.

 Thanks for performing the experiment. The effect in the case of the Thimerosal is not that negligible and should be mentioned. Can the authors perform the same analysis in a later time-point - show a proliferation curve? Moreover, How the authors assess that "only the live cells after thimerosal treatment could activate co-cultured T cells". How were the dead and alive cells separated for the assay?

6) Thanks for the question. We re-analyzed our RNA-seq data and found the top 10 of enriched GO terms in thimerosal-treated cells are associated with cell metabolism, stress, and ROS (Response Figure 19). This is also consistent with our metabolomics screening data.

 The GO analysis should be showed in supplementary since many hypotheses started from this transcriptomic analysis.

7) Thanks for the important suggestions. Our results show that thimerosal triggers tumor immunogenicity via increasing the ROS and activating the RIPK3-IRF1 axis to trigger IRG1 expression and itaconate production in EG7 cells. Indeed, due to the mutation or deficiency of this signaling axis in cancer cells, thimerosal can't induce IRG1 expression and itaconate production in all types of cancer cells. Thus, only those cancer cells with a complete signaling pathway axis may benefit from the thimerosal treatment. We next plan to check the other cell lines which have complete signaling pathway axis, and use the cell lines to verify the IRG1-itaconate axis in tumor cells promoting tumor immunogenicity in vitro and in vivo. We have included the possibility into the revised discussion part.

 Perfect.

8) Thanks for your questions. We have added the detailed information in our revised Figures 3 and 4. Our previous statistical comparisons focused on the difference between the shCTRL and shTFEB. We re-analyzed the data, and the new results showed the 4-OI and thimerosal treatment increased the tumor immunogenicity significantly in shCTRL cells.

 Perfect, all clear now.

9) Fig. 4H: is there any difference upon treatment in the shSCR?

Response: Thanks for your comments and suggestions. We performed statistical analysis on the data, and the 4-OI or thimerosal treatment increased the IFN γ and GzmB expression in the co-cultured OT-I cells significantly. We have added the results in our revised Figure 3H.

 Perfect.

10) Thanks for your comments. The treatment of 4-OI, DI, or thimerosal was removed by refreshing media before co-culture with T cells. We also tested the condition when these treatments were kept in the media throughout the experiment. The results show that presented DI and thimerosal but not 4-OI could still induce co-cultured T cell activation, but weaker than the removed group (Response Figure 20). These data suggested that these drugs mainly act on tumor cells and enhance tumor immunogenicity to induce co-cultured T cell activation.

 Okay, the refreshing step should be added in the material section.

Minor points:

All minor points have been fully addressed.

Few additional minor points:

- 1) The differences in macrophage infiltration (Fig. S8G) should also be mentioned in the text.
- 2) Most of the call to the panels in Fig S4 in the text are wrong.
- 3) In the discussion the authors say "Itaconate could be induced by various signaling pathways in macrophages (13, 17), including the LPSactivated TLR4-NF-kB axis, and the STING/MyD88/IRG1 axis (13, 35)." Please note that also miRs can be involved in the control of itaconate production in activated macrophages as recently showed (PMID: 33962944).

Referee #1:

The authors address my concerns.

Response: Thanks for your encouraging comment.

Referee #2:

The authors have improved their manuscript but there are still some issues.

1. The figure legends still lack sufficient details to interpret them properly e.g. concentrations of drugs.

Response: Thanks for your kind reminders. We have double-checked and included more detailed information in our revised figure legends.

2. Some of the figures that are in the revision file should be included as supplementary data. This includes Response Figure 7, Response Figure 9-11.

Response: Thanks for your insightful suggestions. We have incorporated the Response Figure 7 as our revised "Appendix Fig. S1A-B", the Response Figure 9 as our revised "Appendix Fig. S2A and S2D", the Response Figure 10 as our revised "Appendix Fig. S2F", and the Response Figure 11 as our revised "Figure 2J and Appendix Fig. S2K"

3. The revised in vivo data improves the manuscript.

Response: Thanks for your encouraging comment.

Referee #3:

The paper by Wang et al describes how the increase of IRG1, key enzyme for itaconate metabolism, can boost the antigenicity of cancer cells leading to a better anti-tumoral T cell response both in vitro and in vivo. Moreover, they described how thimerosal, a vaccine preservative, can be use to therapeutically induce IRG1 specifically in cancer cell and how this treatment can synergize with immunotherapy.

Although growing literature describes the role of itaconate in the tumor microenvironment, the role of IRG1 in cancer cells is still almost totally elusive. For this reason, combined with the strong efficacy of thimerosal in reducing tumor growth, the reviewer believes that the data are interesting and novel in

the field of cancer immunology and cancer therapy.

Moreover, compared with the previous version of the manuscript assessed by the present Reviewer there is a better elucidation of the role of itaconate vs its derivatives. In parallel, additional experiments provided further clues to understand why the treatment with thimerosal induces Irg1 specifically in cancer cells and not in macrophages. Finally, a better description of the molecular pathway induced by thimerosal treatment and additional in vivo analysis have been performed.

The paper is well-written and the experiments are generally well performed and the message of the manuscript is strengthened in the current version. Despite this, the Reviewer believe that few minor comments need to be addressed before acceptance in EMBO Journal:

1) Response: Thanks for your encouraging comments and suggestions. We sorted out the immune cells and non-immune cells from the PBS- or thimerosal-treated tumor tissues and examined IRG1 expression by WB. The result shows that thimerosal treatment leads to significant upregulation of IRG1 expression in non-immune cells but not immune cells (Response Figure 15). Our results show that thimerosal could trigger tumor immunogenicity in vitro and in vivo. Moreover, due to the upregulated immunogenicity induced by thimerosal treatment, more CD8+ T cells were infiltrated into tumor tissue and activated, which in turn inhibits tumor growth, activates tumor microenvironment, and boosts the host's anti-tumor immune response circle. We think this may be the reason why only one injection of thimerosal could be sufficient to inhibit tumor growth in a relatively long term. We have incorporated the data in Response Figure 15 into revised Figure S8B.

 The data are very convincing (although the quality of the WB for the immune cells should be improved). Despite this, some information is missing, in particular which one was the gating strategy to sort the cells? Did the authors use CD45+ vs CD45-?

Response: Thanks for your encouraging comment and kind suggestions. Following your suggestion, we repeated the WB to check the IRG1 expression in the immune cell from PBS or thimerosal-treated tumor tissue. Consistent with our previous result, thimerosal treatment doesn't influence the expression of IRG1 in immune cells. In addition, the quality of the WB was improved, and have incorporated the data into our revised "Appendix Figure S8B".

We apologize for skipping the information on the gating strategy to sort the cells. Yes, we sorted the cells by CD45 expression and considered the CD45+ population as immune cells and the CD45- population as non-immune cells. And we have incorporated the information into our revised "Appendix Figure S8 legend".

2) Thanks for your encouraging comments and important suggestions. We have added the data of itaconate levels intracellular vs extracellular in our revised

manuscript (Figure S2), and we used the medium supplemented with itaconate as the positive control (PC) (Response Figure 16A). We also checked the IRG1 expression and itaconate production in EG7 cells at different time points after thimerosal treatment. The results show IRG1 and itaconate are continuously accumulated in EG7 cells (Response Figure 16B-C).

 The data are convincing. Despite this, I would perform a later time-point such as 48 or 72 hours to check both intracellular and, more importantly, extracellular levels to be sure that itaconate is not exported when it started to reach relatively "high intracellular levels"

Moreover, based on the importance of extracellular itaconate, the Reviewer would suggest to put the panel A in the main figure.

Response: Thanks for your encouraging comments and kind suggestions. As you suggested, we checked the expression of IRG1 in EG7 cells which were treated with thimerosal (10 μ M) for 24, 48, or 72 hrs. Our results showed that IRG1 protein was expressed at the highest level after 24 hrs of thimerosal treatment (**Response Figure 1A**). Moreover, thimerosal treatment for 48 or 72 hrs would induce a large amount of cell death, which may be due to the sustained stress or long-term cell cycle arrest (**Response Figure 1B**). In addition, thimerosal treatment did not induce the expression level of ABCG2, a recently identified itaconate exporter (*Chen et al., Cell Metab, 2024; PMID: 38181789*), in EG7 cells (**Response Figure 1C**). Taken together, these data suggest thimerosal induces itaconate production in the early time point but triggers cell death in long-term treatment, and the accumulated itaconate would unlikely get exported after long-term thimerosal treatment. As you suggested, we have incorporated panel A into our revised "Figure 2E".

Figure for reviewers removed

3) Response: Thanks for your question. Our result shows overexpression of TFEB didn't increase the B2m and LacZ activity as the treatment of 4-OI or thimerosal. We also noted that overexpressed TFEB was mainly in the cytoplasmic fraction, and thimerosal treatment enhanced its nuclear translocation (Response Figure 5E). Thus, overexpressed TFEB still needs

further stimulation for nuclear translocation and gene induction. We speculate that without stimulation, most of the overexpressed TFEB would be likely to be phosphorylated by mTOR and bind with 14-3-3 to keep their cytosolic retention (6, 13). Once the cells were treated with 4-OI or thimerosal, the TFEB would be modified by 4-OI or itaconate and then translocated into nuclear and led gene expression. Future studies would be needed to delineate the detailed mechanism.

 Thank you for the answer, the proposed speculation sounds reasonable.

Response: Thanks for your encouraging comment.

4) Thanks for the great suggestion. We used an increased dosage of thimerosal to treat EG7 cells and found that thimerosal indeed induced IRG1 expression in a dosedependent fashion (Response Figure 17).

 Thank you, very convincing data.

Response: Thanks for your encouraging comment.

5) Thanks for your great suggestions. Following your suggestion, we used thimerosal or 4-OI treated with EG7 cells for 18 hrs and then checked the cell cycle (via PI staining) and apoptosis (via apoptosis assay kit) of EG7 cells by FACS. The results showed that thimerosal but not 4-OI increased G0/G1 population (Response Figure 18A), and thimerosal and 4-OI both induced very few cell apoptosis (Response Figure 18B-C). Moreover, we also found that only the live cells after thimerosal treatment could activate co-cultures T cells (Response Figure 18D). Taken together, we conclude that thimerosal and 4-OI trigger tumor immunogenicity independent of the influence of cell cycle and cell apoptosis.

 Thanks for performing the experiment. The effect in the case of the Thimerosal is not that negligible and should be mentioned. Can the authors perform the same analysis in a later time-point - show a proliferation curve? Moreover, How the authors asses that "only the live cells after thimerosal treatment could activate co-cultures T cells". How were the dead and alive cells separated for the assay?

Response: Thanks for your encouraging comment and kind suggestions. Following your suggestions, we measured cell proliferation of EG7 cell after thimerosal or 4-OI treatment, and the results showed that thimerosal induced notable cell proliferation inhibition and 4-OI shows partial cell proliferation inhibition (Response Figure 2). It is likely that the long-time treatment of these drugs would arrest the cell cycle or induce sustainable stress, and in turn inhibit cell proliferation and then initiate cell death.

To assess the T cell-activating effect of live or dead cells, we stained thimerosal-treated EG7 cells with Zombie Aqua™ dye (BioLegend, 42301), and then

sorted the zombie-positive cells (Dead cells) or the zombie-negative cells (Live cells) via flow cytometry, then used live or dead cells for co-culture with B3Z cells.

Figure for reviewers removed

6) Thanks for the question. We re-analyzed our RNA-seq data and found the top 10 of enriched GO terms in thimerosal-treated cells are associated with cell metabolism, stress, and ROS (Response Figure 19). This is also consistent with our metabolomics screening data.

 The GO analysis should be showed in supplementary since many hypotheses started from this transcriptomic analysis.

Response: Thanks for your encouraging comment and kind suggestion. We have incorporated the result as our revised "Appendix Fig. S7A"

7) Thanks for the important suggestions. Our results show that thimerosal triggers tumor immunogenicity via increasing the ROS and activating the RIPK3-IRF1 axis to trigger IRG1 expression and itaconate production in EG7 cells. Indeed, due to the mutation or deficiency of this signaling axis in cancer cells, thimerosal can't induce IRG1 expression and itaconate production in all types of cancer cells. Thus, only those cancer cells with a complete signaling pathway axis may benefit from the thimerosal treatment. We next plan to check the other cell lines which have complete signaling pathway axis, and use the cell lines to verify the IRG1-itaconate axis in tumor cells promoting tumor immunogenicity in vitro and in vivo. We have included the possibility into the revised discussion part.

 Perfect.

Response: Thank you for your encouraging comment.

8) Thanks for your questions. We have added the detailed information in our revised Figures 3 and 4. Our previous statistic comparisons focused on the difference between the shCTRL and shTFEB. We re-analyzed the data, and the new results showed the 4-OI and thimerosal treatment increased the tumor immunogenicity significantly in shCTRL cells.

 Perfect, all clear now.

Response: Thanks for your encouraging comment.

9) Fig. 4H: is there any difference upon treatment in the shSCR?

Response: Thanks for your comments and suggestions. We performed statistical analysis on the data, and the 4-OI or thimerosal treatment increased the IFN γ and GzmB expression in the co-cultured OT-I cells significantly. We have added the results in our revised Figure 3H.

 Perfect.

Response: Thanks for your encouraging comment.

10) Thanks for your comments. The treatment of 4-OI, DI, or thimerosal was removed by refreshing media before co-culture with T cells. We also tested the condition when these treatments were kept in the media throughout the experiment. The results show that presented DI and thimerosal but not 4-OI could still induce co-cultured T cell activation, but weaker than the removed group (Response Figure 20). These data suggested that these drugs mainly act on tumor cells and enhance tumor immunogenicity to induce co-cultured T cell activation.

 Okay, the refreshing step should be added in the material section.

Response: Thanks for your encouraging comment and kind suggestion. We have included the detail of the experiment condition in our revised "Method" section.

Minor points:

All minor points have been fully addressed.

Response: Thanks for your encouraging comment

.

Few additional minor points:

1) The differences in macrophage infiltration (Fig. S8G) should also be mentioned in the text.

Response: Thanks for your kind suggestion. We have described this result in our revised "Result" section.

2) Most of the call to the panels in Fig S4 in the text are wrong.

Response: Thanks for your kind reminder. We have corrected the mistakes in our revised text.

3) In the discussion the authors say "Itaconate could be induced by various signaling pathways in macrophages (13, 17), including the LPS activated TLR4-NF- κ B axis, and the STING/MyD88/IRG1 axis (13, 35)." Please note that also miRs can be involved in the control of itaconate production in activated macrophages as recently showed (PMID: 33962944).

Response: Thanks for your kind reminder. We have now cited and briefly discussed the findings in this reference in our revised manuscript. (Reference: *Virga, et al., Sci Adv, 2021*)

Dear Dr Xia,

Thank you for submitting the revised version of your manuscript. I have now evaluated your amended manuscript and concluded that the remaining minor concerns by the referees have been sufficiently addressed.

I am thus pleased to inform you that your manuscript has been accepted for publication in the EMBO Journal.

Also, we kindly ask for your consent on keeping the referee figures included in this file.

On a different note, I would like to alert you that EMBO Press offers a format for a video-synopsis of work published with us, which essentially is a short, author-generated film explaining the core findings in hand drawings, and, as we believe, can be very useful to increase visibility of the work. Please see the following link for representative examples and their integration into the article web page:

<https://www.embopress.org/doi/full/10.15252/emj.2019103932>

If you have any questions, please do not hesitate to contact the Editorial Office.

Kind regards,

Daniel Klimmeck

Daniel Klimmeck, PhD
Senior Editor
The EMBO Journal
EMBO
Postfach 1022-40
Meyerhofstrasse 1
D-69117 Heidelberg
contact@embojournal.org